# The kinase ZYG-1 phosphorylates the cartwheel protein SAS-5 to drive centriole assembly in *C. elegans*

Prabhu Sankaralingam [1✉], Shaohe Wang [2], Yan Liu[1], Karen F Oegema[3,4] & Kevin F O'Connell [1✉]

## Abstract

**Centrioles organize centrosomes, the cell's primary microtubule-organizing centers (MTOCs). Centrioles double in number each cell cycle, and mis-regulation of this process is linked to diseases such as cancer and microcephaly. In *C. elegans*, centriole assembly is controlled by the Plk4 related-kinase ZYG-1, which recruits the SAS-5–SAS-6 complex. While the kinase activity of ZYG-1 is required for centriole assembly, how it functions has not been established. Here we report that ZYG-1 physically interacts with and phosphorylates SAS-5 on 17 conserved serine and threonine residues in vitro. Mutational scanning reveals that serine 10 and serines 331/338/340 are indispensable for proper centriole assembly. Embryos expressing SAS-5[S10A] exhibit centriole assembly failure, while those expressing SAS-5[S331/338/340A] possess extra centrioles. We show that in the absence of serine 10 phosphorylation, the SAS-5–SAS-6 complex is recruited to centrioles, but is not stably incorporated, possibly due to a failure to coordinately recruit the microtubule-binding protein SAS-4. Our work defines the critical role of phosphorylation during centriole assembly and reveals that ZYG-1 might play a role in preventing the formation of excess centrioles.**

**Keywords** Centriole; Kinase; Protein–Protein Interaction; *C. elegans*
**Subject Category** Cell Adhesion, Polarity & Cytoskeleton

## Introduction

Centrioles are submicron-scale cylindrically shaped organelles, with a central cartwheel and an outer wall composed of a ninefold symmetric arrangement of microtubules (Gonczy, 2012; Winey and O'Toole, 2014). They are evolutionarily conserved organelles present in all eukaryotes with the exception of most fungi and higher plants (Carvalho-Santos et al, 2010; Hodges et al, 2010;

Marshall, 2009; Ross, 1968; Schwarz et al, 2018). In cycling cells, centrioles recruit a dense proteinaceous material called the pericentriolar material or PCM, thereby forming a centrosome, the cell's primary microtubule-organizing center or MTOC. By nucleating and anchoring arrays of microtubules, centrosomes help establish the poles of the mitotic spindle, thereby promoting the proper segregation of chromosomes. In quiescent and terminally differentiated cells, the centrioles act as basal bodies to template cilia that may have a mechanical or signaling function (Arquint et al, 2014; Woodruff et al, 2014).

Forward genetic and RNAi-based screens in the nematode *Caenorhabditis elegans* have been key to identifying a set of core centriolar proteins—the serine/threonine kinase ZYG-1 (O'Connell et al, 2001), and the coiled-coil proteins—SPD-7 (Sugioka et al, 2017), SPD-2 (Kemp et al, 2004; Pelletier et al, 2004), SAS-5 (Dammermann et al, 2004; Delattre et al, 2004), SAS-6 (Leidel et al, 2005) and SAS-4 (Kirkham et al, 2003; Leidel and Gonczy, 2003). The homologs of many of these proteins are found across genera, including humans and flies (Carvalho-Santos et al, 2010; Hodges et al, 2010; Marshall, 2009; Ross, 1968; Schwarz et al, 2018). Sak (Bettencourt-Dias et al, 2005) and Plk4 (Habedanck et al, 2005) are *Drosophila* and human orthologs of ZYG-1, respectively. Worm SAS-5 is distantly related to human STIL (Tang et al, 2011; Vulprecht et al, 2012) and *Drosophila* Ana2 (Stevens et al, 2010), while SAS-4 has homologs in flies and humans (CPAP) (Basto et al, 2006; Kirkham et al, 2003; Leidel and Gonczy, 2003).

In dividing cells, centriole duplication is coupled to DNA replication and happens only once per cell cycle to ensure that two pairs of centrioles are present at mitosis (Loncarek et al, 2008; Nigg, 2007; Tsou and Stearns, 2006). Mis-regulation of centriole number leads to mono and multipolar spindles, errors in chromosome segregation, and ultimately cell division failure. Not surprisingly, such defects have been associated with a variety of human disorders, including primary microcephaly (Marthiens et al, 2013) and cancer (Basto et al, 2008; Sabino et al, 2015).

Electron microscopic studies of centrioles and basal bodies have been key to elucidating their architecture. The striking feature of a centriole viewed in cross section is the presence of a "cartwheel"-like structure in the lumen. The cartwheel possesses a central hub

[1]Laboratory of Biochemistry and Genetics, National Institutes of Diabetes and Digestive and Kidney Diseases, NIH, Bethesda, MD, USA. [2]Janelia Research Campus, Howard Hughes Medical Institute, Ashburn, VA, USA. [3]Department of Cell and Developmental Biology, School of Biological Sciences, University of California, San Diego, La Jolla, CA 92093, USA. [4]Department of Cellular and Molecular Medicine, University of California San Diego, La Jolla, CA 92093, USA. ✉E-mail: prabhu.sankaralingam@nih.gov; kevino@nih.gov

and nine radiating spokes that attach to sets of microtubules at the periphery (Guichard et al, 2013; Li et al, 2012). Nematode centrioles were originally thought to lack a cartwheel as EM revealed only a relatively simple central tube within the lumen (Pelletier et al, 2006). However, a recent study utilizing expansion microscopy showed that *C. elegans* centrioles indeed contain such a cartwheel-like structure at their core (Woglar et al, 2022).

Genetic and ultrastructural studies in *C. elegans* have delineated the hierarchical requirements of the core centriole proteins. SAS-7 is at the apex of the assembly cascade and recruits SPD-2 to the site of procentriole assembly on the pre-existing centriole (Sugioka et al, 2017). SPD-2 in turn localizes the master regulatory kinase ZYG-1 (Dammermann et al, 2004; Pelletier et al, 2004; Shimanovskaya et al, 2014). A complex of SAS-6 and SAS-5 is then recruited through direct interaction of SAS-6 with ZYG-1 (Delattre et al, 2006; Lettman et al, 2013; Pelletier et al, 2006), forming a nascent central tube (or cartwheel), which is ultimately stabilized by the addition of SAS-4 and microtubules to the outer wall (Pelletier et al, 2006).

Formation of the cartwheel is critical for centriole assembly. The protein SAS-6 has been shown to be the key structural component of the cartwheel, which imparts the ninefold symmetry to centrioles (Kitagawa et al, 2011b; Nakazawa et al, 2007; Woglar et al, 2022). Fascinatingly, at high concentrations, recombinant SAS-6 can self-assemble in vitro into planar ring-like structures resembling centrioles (Gopalakrishnan et al, 2010; Guichard et al, 2017; Hilbert et al, 2016; Kitagawa et al, 2011b). The crystal structure of truncated SAS-6 shows that it is an elongated molecule, with a globular head domain and an extended α-helical tail. SAS-6 exists as a dimer in solution, with dimerization occurring through the extended coiled coil of the tail. The dimer is the building block for higher-order oligomerization mediated by the interaction of head domains which form the hub, while the extended coiled-coil tails form the spokes. However, the interaction of head domains is quite weak ($K_d$ ~50–100 μM) (Kitagawa et al, 2011b; van Breugel et al, 2011; van Breugel et al, 2014), indicating that other core centriolar proteins may act to stabilize SAS-6 at centrosomes during cartwheel assembly.

Studies in *C. elegans* indicate that cartwheel assembly occurs in two separable steps in vivo, where recruitment of SAS-6 to the nascent centriole and its stable incorporation are distinct events (Lettman et al, 2013). The kinase activity of ZYG-1 is dispensable for localizing SAS-6 to the assembly site but is absolutely required to form a stable cartwheel (Lettman et al, 2013). This indicates that ZYG-1 phosphorylates yet unknown substrates during cartwheel assembly. While SAS-6 has been shown to be a substrate of ZYG-1 (Kitagawa et al, 2009), phosphorylation of SAS-6 is not necessary for cartwheel formation (Lettman et al, 2013). Thus the critical target of ZYG-1 remains unknown.

In this study, we identify SAS-5 residues phosphorylated by ZYG-1 in vitro and show that one of these near the N-terminus is necessary for centriole assembly in vivo. Lack of phosphorylation at this key residue causes failure of SAS-4 recruitment and destabilization of the growing cartwheel. We also identify a set of three serine residues at the C-terminus that appear to play a role in maintaining proper levels of SAS-5 and ensuring the production of one and only one daughter centriole. Collectively, our data indicate that phosphorylation of SAS-5 serves as a regulatory hub for proper control of centriole assembly.

# Results

## Recombinant expression, purification, and in vitro refolding of core centriolar proteins

One of the major impediments in studying the core centriolar components is the difficulty in heterologous recombinant protein expression in hosts such as *Escherichia coli*. The kinase ZYG-1 is toxic to bacteria when expressed in soluble form, needing high volumes of culture to obtain appreciable quantities (Cottee et al, 2013). The other coiled-coil components such as SAS-5 (Rogala et al, 2015), SAS-6 and SAS-4 (Gopalakrishnan et al, 2011) are expressed in the insoluble fraction. Much of the earlier efforts to obtain soluble recombinant proteins involved using solubilizing protein tags or heterologous expression in cell lines. These methods suffer from issues such as long incubation times with proteases to remove the tags and protein aggregation following tag removal. Posttranslational modifications and co-eluting host proteins when expressed in cell lines hinder downstream experiments. Alternative approaches have used truncated proteins or protein fragments, but the properties of such proteins may not accurately reflect the function of full-length proteins. To circumvent these issues, we purified full-length proteins from bacterial inclusion bodies and refolded them in vitro. All proteins were expressed as C-terminal His$_6$ fusions and were purified under denaturing conditions by capturing them on IMAC (Immobilized Metal Affinity Chromatography) resin (Fig. 1A). The homogeneity of the refolded proteins was assessed by SDS-Polyacrylamide Gel Electrophoresis (SDS-PAGE) (Fig. 1B) and mass spectrometry, which only detected a few peptides derived from *E. coli* proteins, some of which are most commonly found in inclusion bodies. These peptides represented only 0.3–0.07% of the sequence coverage. Note that kinase active ZYG-1::His$_6$ runs as a broad unfocused band (Fig. 1B) as observed in other studies (Kratz et al, 2015; Lettman et al, 2013). For the sake of clarity, in the text that follows, we will omit the use of the His$_6$ suffix.

To assess whether the refolded proteins attained their native functional conformation, we performed circular dichroism spectropolarimetry (CD) measurements (Fig. EV1). The CD spectra of refolded ZYG-1, SAS-6, and SAS-5 confirmed the presence of secondary structural elements. The spectrum for ZYG-1 indicated the existence of a mixture of α-helix and β-sheets, consistent with the established structure of kinase domains and the crystal structure of the cryptic polo box (CPB) of ZYG-1 (Shimanovskaya et al, 2014). Further, as shown in Fig. 1C, the in vitro refolded ZYG-1 exhibited enzymatic activity. For SAS-6, its partial crystal structure has been solved (Hilbert et al, 2013; Qiao et al, 2012) revealing an N-terminal globular head domain and an elongated α-helical C-terminal domain (van Breugel et al, 2011; van Breugel et al, 2014). Consistent with its known structure, our CD spectrum of SAS-6 shows that it has a large α-helical content (Fig. EV1). Finally, an earlier study of near full-length SAS-5 by Rogala et al, provided us with an opportunity to compare the CD spectrum of our refolded SAS-5 with that purified in soluble form. We found that our CD spectrum of full-length refolded SAS-5 within the wavelength range of 200–250 nm is nearly identical with that obtained by the previous study (Rogala et al, 2015) (Fig. EV1). In contrast, the CD spectrum of full-length SAS-4, resembles that of natively unfolded proteins (Ahmad et al, 2006; Weinreb et al, 1996)

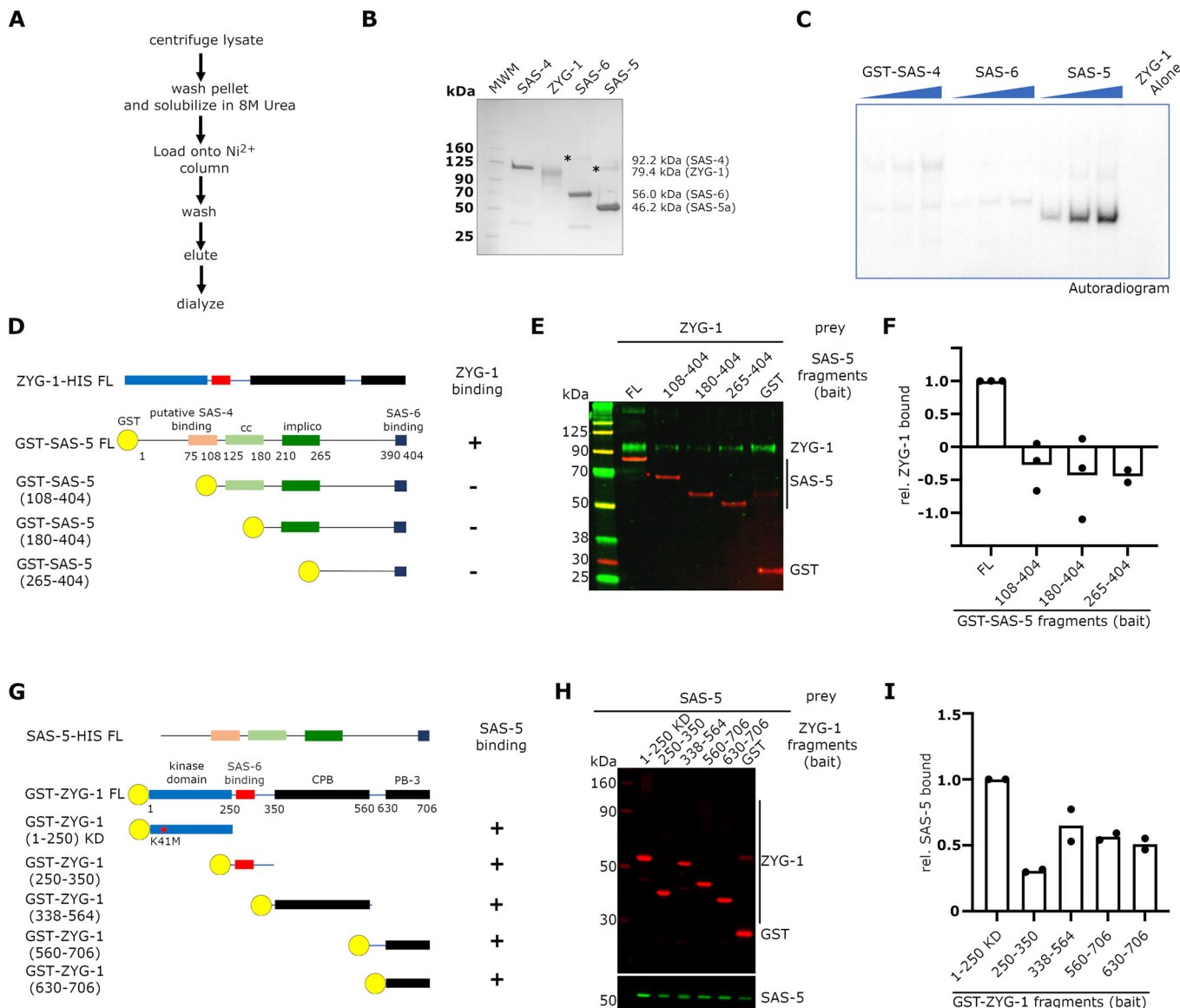

**Figure 1. ZYG-1 binds SAS-5 and phosphorylates it in vitro.**

(A) Scheme used for the purification and refolding of his-tagged proteins from inclusion bodies. Prior to the final step, all buffers contained 8 M urea. (B) In vitro refolded proteins were separated on 4–12% NuPAGE gel under denaturing conditions and stained with Coomassie blue. Asterisks indicate disulphide-linked dimers. The theoretical masses of the proteins are indicated. Note that only the longer SAS-5a isoform was used in these studies. (C) Autoradiogram of an in vitro kinase assay employing 60 nM ZYG-1 and purified GST-SAS-4, SAS-6, and SAS-5. The wedges above the autoradiogram indicate increasing amounts of substrate (1, 2, and 3 µM). While ZYG-1 phosphorylates all three centriole proteins, SAS-5 is phosphorylated to a higher degree. (D) Schematic depicting the full-length ZYG-1 and GST-SAS-5 protein fragments used for pull-down assays. For each SAS-5 protein, the ability to bind ZYG-1 is indicated. (E) Image of an immunoblot showing that full-length SAS-5, but not the N-terminally truncated versions of SAS-5, can pull down ZYG-1. (F) Quantitation of ZYG-1 binding assays. The bar graph indicates the average value of independent biological replicates. Note that for each SAS-5 construct, binding is detected only when the amount of ZYG-1 captured is greater than that observed for the negative control (GST alone). (G) Schematic depicting the full-length SAS-5 and GST-ZYG-1 protein fragments used for pull-down assays. For each ZYG-1 protein fragment, the ability to bind SAS-5 is indicated. (H) Image of an immunoblot showing that full-length SAS-5 can interact with all the GST-ZYG-1 fragments. (I) Quantitation of SAS-5 binding assays. The bar graph indicates the average value of independent biological replicates. Note that for each SAS-5 protein fragment, binding is detected only when the amount of ZYG-1 captured exceeds that observed for the negative control (GST alone). The K41M mutation was used in the GST-ZYG-1(1-250) protein to reduce its toxicity to E. coli. FL full-length, KD kinase-dead. Source data are available online for this figure.

(Fig. EV1). The was surprising because the crystal structure of the C-terminal domain of *Danio rerio* and *Drosophila* SAS-4, known as the G-box or the TCP (T complex protein 10) domain, has been shown to consist of cross β-sheets reminiscent of amyloid fibrils (Cottee et al, 2013; Hatzopoulos et al, 2013). However, the structure of *C. elegans* SAS-4 has not yet been investigated, and thus while it is possible that under the conditions of our purification strategy SAS-4 did not refold into its native conformation, it is also possible that *C. elegans* SAS-4 may only attain a mature structure upon interaction with other proteins, as shown with other natively

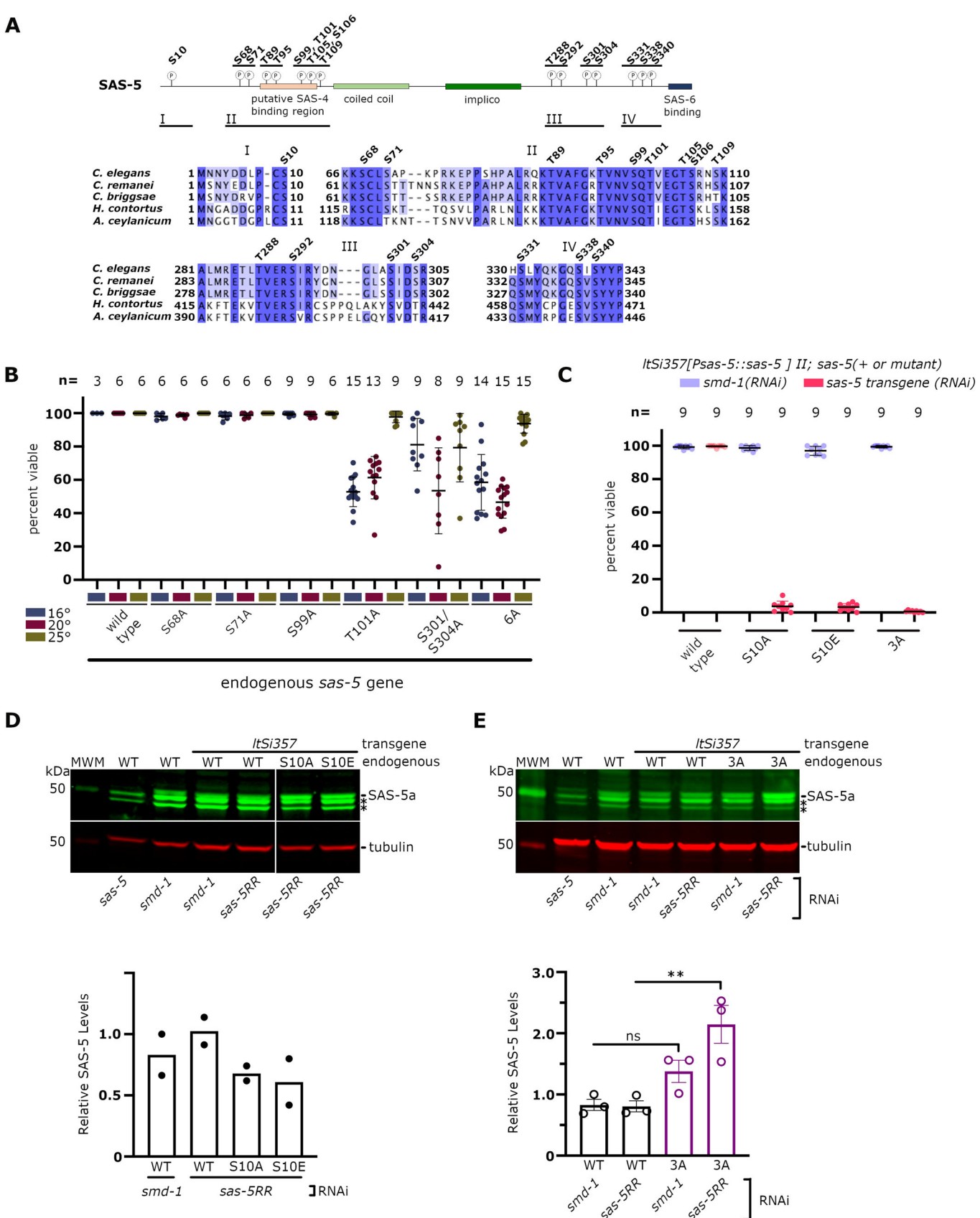

◄ **Figure 2. ZYG-1 phosphorylates SAS-5 on several conserved serine and threonine residues that are required for SAS-5 function in vivo.**

(A) Phosphoproteomic analysis identified 37 serine and threonine residues that are phosphorylated by ZYG-1 in vitro. Seventeen of these residues are conserved broadly among nematodes (shown in bold above schematic and above sequence) and are located in four separate blocks of homology (I–IV). Note that SAS-5 is phosphorylated all along its length. (B) Embryonic viability of strains homozygous for the indicated mis-sense mutations in the endogenous sas-5 gene. Each strain was tested at 16°, 20°, and 25 °C. The mean and standard error of the mean (SEM) are shown, with each point representing the percent of viable progeny of a single hermaphrodite. (C) The three mutants (S10A, S10E, and 3A) that are homozygous sterile were maintained in the presence of a wild-type recoded sas-5 transgene (ltSi357). Embryonic lethality was measured in worms subjected to control (smd-1) RNAi or RNAi specifically targeted to silence the transgene at 20 °C. The mean and SEM are shown, with each point representing the percent of viable progeny of a single hermaphrodite. (D) Representative immunoblot showing that expression of the S10A and S10E mutants are comparable to the wild-type SAS-5 protein upon RNAi against the transgene (sas-5RR). Asterisks indicate two nonspecific bands. The quantitation of SAS-5 levels is shown below. Note that silencing of the transgene (sas-5RR) did not have an effect on expression of the wild-type SAS-5 protein (compare WT/smd-1 to WT/sas-5RR). Note also that the levels of S10A and S10E are similar to those of the wild type. (E) Quantitative immunoblot showing that expression of the 3A mutant is elevated relative to the wild-type SAS-5 protein. Asterisks indicated two nonspecific bands. The quantitation of SAS-5 levels is shown below ($n = 3$). The mean and SEM are shown. ns not significant. **$P < 0.01$ by one-way ANOVA. Source data are available online for this figure.

unfolded proteins (Onitsuka et al, 2008; Sugase et al, 2007). In summary, based on the CD spectra, the acquisition of ZYG-1 enzymatic activity, and the specific protein–protein interactions demonstrated below, we believe our purification strategy yields full-length natively folded proteins that behave in a physiologically relevant manner.

## ZYG-1 directly binds SAS-5 and preferentially phosphorylates it in vitro

The in vitro refolded full-length ZYG-1 was used in a kinase assay to investigate its ability to phosphorylate core centriolar components. To avoid nonspecific phosphorylation of substrates, we used nanomolar concentrations of ZYG-1. In this assay, SAS-6 served as a positive control since it had previously been shown to be a substrate of ZYG-1 in vitro (Kitagawa et al, 2009; Lettman et al, 2013). As shown in Fig. 1C, we found that ZYG-1 can indeed phosphorylate SAS-6, as well as GST-SAS-4, and SAS-5. Interestingly, we found that in comparison to SAS-4 and SAS-6, SAS-5 is extensively phosphorylated by ZYG-1. This observation prompted us to focus on SAS-5 and determine how it interacts with ZYG-1. We therefore performed in vitro pull-down experiments with GST-fusions of full-length and N-terminal truncations of SAS-5 as bait to capture ZYG-1 (prey) from solution. We found that full-length GST-SAS-5 could interact with ZYG-1 (Fig. 1D–F). However, we failed to detect binding above background for all N-terminally truncated version of SAS-5, include one lacking just the first 107 amino acids. It is possible that these N-terminally truncated proteins bind to ZYG-1 with reduced efficiency, but that such weak binding is not detectable due to the relatively high background of our assay. Nonetheless, it should be noted that the protein fragment spanning amino acids 108–404 retains the coiled-coil (CC) domain that has been shown to mediate binding to Plk4 in vertebrates and flies (Arquint et al, 2015; Cottee et al, 2017; McLamarrah et al, 2018; Ohta et al, 2014). This truncated protein, however, fails to exhibit detectable binding to ZYG-1, suggesting that SAS-5 interacts with ZYG-1 via a distinct mechanism.

To determine which regions within ZYG-1 mediate binding to SAS-5, we expressed and purified several fragments that spanned the entire length of ZYG-1 as GST fusion proteins (Fig. 1G). These were then used as bait to pull down full-length SAS-5. As shown in Fig. 1H,I, multiple regions of ZYG-1 exhibit binding to SAS-5. In particular, the kinase domain (amino acids 1–250) showed the strongest binding. While it remains to be seen if these later

interactions contribute significantly to ZYG-1-SAS-5 binding, similar results have been obtained in vertebrates and flies (Cottee et al, 2017; Kratz et al, 2015; McLamarrah et al, 2018; Ohta et al, 2014).

## ZYG-1 phosphorylates multiple conserved serine and threonine residues in SAS-5 in vitro

We next sought to identify the residues in SAS-5 phosphorylated by ZYG-1. We thus performed an in vitro kinase assay with ZYG-1 and analyzed phosphorylated SAS-5 by mass spectrometry. We found that ZYG-1 could phosphorylate SAS-5 at serine and threonine residues positioned throughout the length of the protein (Fig. 2A). PEAKS software was used to analyze the mass spectrometry data and to calculate an Ambiguity Score (AScore), which reflects the confidence with which peptides are identified (Han et al, 2011). We obtained sequence coverage of 94% and 62% for SAS-5 and ZYG-1, respectively. The high sequence coverage decreased the probability of missing any phosphorylation event. Applying an AScore cut-off of 18, we identified 146 peptides comprising a total of 37 distinct phosphorylated serine and threonine residues (Dataset EV1). To determine which of these sites might play an important physiological role, we compared the sequences of SAS-5 from C. elegans and distantly related nematodes. While overall the nematode SAS-5 proteins have diverged extensively, they share four tracts with high sequence identity (Fig. 2A). Specifically, the regions spanning residues 1–10, 66–110, 281–305, and 330–343 in C. elegans SAS-5 are highly homologous with respect to position and sequence with all of the other species examined. We found that 17 of the 37 serine and threonine residues that were phosphorylated by ZYG-1 in vitro were conserved among nematodes and that all were contained within one of the four tracts of conservation (Fig. 2A).

## Mutational screening reveals four conserved serine residues are absolutely required for sas-5 function

To address if any of the 17 conserved serine and threonine residues identified as ZYG-1 targets in vitro are important for SAS-5 function in vivo, we took two complementary approaches. First, we generated a set of RNAi-resistant single-copy transgenes under the control of endogenous sas-5 regulatory sequences integrated into a specific site on chromosome II (Fig. EV2A,B). The transgenes directed the expression of WT SAS-5 (transgene ltSi357) or SAS-5 with specific sets of serine/threonine residues mutated to alanine

(Fig. EV2C). This set includes all 17 conserved residues plus a few additional serines and threonines. As shown in Fig. EV2D, RNAi of the endogenous *sas-5* gene produces 100% embryonic lethality in a control strain lacking the recoded *sas-5* transgene, but no phenotype in a strain carrying the wild-type re-encoded transgene. This indicates that our RNAi protocol specifically targets the endogenous gene and that the re-encoded transgene is functional. Of the six clusters tested, only two exhibit a strong embryonic lethal phenotype upon RNAi (Fig. EV2D): cluster I comprising serines S10, S68, and S71 and cluster VI comprising serines S331, S338, and S340.

We next determined which of the individual mutations within each cluster contributed to the embryonic lethal phenotypes by constructing single and double mutants (Fig. EV2E,F). As shown in Fig. EV2D, mutation of serine 10 to alanine results in 100% embryonic lethality while no phenotype is observed when S68 and S71 are individually mutated to alanine. Thus, serine 10 appears to be a critical target of ZYG-1. However, combining the S68A and S71A mutations produces a moderate embryonic lethal phenotype (30% viable), suggesting that phosphorylation of these two residues might contribute to SAS-5 function. Interestingly, each of the serines (S331, S338, and S340) within cluster VI produces a fully penetrant embryonic lethal phenotype when mutated individually (Fig. EV2F), suggesting that all could serves as major sites of regulation. Overall, this approach identified four serine residues as important: S10, S331, S338, and S340.

As a second approach to screen serine and threonine residues, we used CRISPR-Cas9 genome editing to target the endogenous locus, converting 11 of the residues to alanine either individually or in various combinations. Mutation of eight residues were found to have either no effect or a somewhat moderate effect on SAS-5 function (Fig. 2B). Conversion of serines 68, 71, or 99 to alanine had no effect on embryonic viability across a range of temperatures indicating that phosphorylation of any of these residues is not essential for SAS-5 function. In contrast, conversion of threonine 101 to alanine results in a partial embryonic lethal phenotype that is temperature dependent; when grown at 25 °C, the SAS-5$^{T101A}$ mutant exhibits nearly wild-type levels of embryonic viability, but at lower temperatures, it produces only 50–60% viable offspring. Conversion of both serines 301 and 304 to alanine also results in a partial embryonic lethal phenotype whereby 50–80% of the progeny survive. Finally, we created a mutant in which six residues (S99, T101, T105, S106, S301, and S304) were converted to alanine. This mutant, which we refer to as 6A displays a partial embryonic lethal phenotype comparable in magnitude to that of the T101A mutant. We conclude that while these six residues might be phosphorylated to facilitate SAS-5 function, they are not essential phosphorylation targets in vivo.

We also targeted the four residues that we had found to be critical using mutant versions of the transgene. Mutation of serine 10 to alanine results in homozygotes that are both sterile and uncoordinated. This phenotype is often associated with strong loss-of-function mutations in essential cell division genes. In such cases, the embryonic divisions of homozygous mutants are supported by the maternal contribution of their heterozygous mothers and subsequent failure of postembryonic cell divisions leads to the generation of sterile uncoordinated adults (Albertson, 1984) (O'Connell et al, 1998). This suggests that SAS-5$^{S10A}$ might be a complete or near complete loss-of-function allele. Indeed, during

the course of our studies, we produced a likely null allele of *sas-5* in which a frameshift mutation truncates the protein after the first 100 amino acids. Animals homozygous for this allele, named *sas-5(bs267)*, exhibited a sterile uncoordinated phenotype that was indistinguishable from that of the SAS-5$^{S10A}$ mutant. We conclude that conversion of serine 10 to alanine strongly, if not completely, abrogates SAS-5 function. Similarly, conversion of the three clustered serine residues near the C-terminus (serines 331, 338, and 340) also resulted in a recessive sterile uncoordinated phenotype. Curiously, we could not recover animals homozygous for single serine-to-alanine mutations at positions 331, 338, or 340. All such mutants were sterile as heterozygotes.

Overall, both approaches that we used to screen conserved residues identified serines 10, 331, 338, and 340 as the most critical for SAS-5 function. However, we noticed that mutations in the transgene tended to be milder than mutations in the endogenous locus. For instance, we observe an embryonic lethal phenotype when the T101A mutation is present in the endogenous locus (Fig. 2B) but no phenotype when it is present in the transgene (cluster II, Fig. EV2D). Likewise, the S10A and S331/S338/S340A mutants present as essentially null alleles (sterile and uncoordinated phenotype) when present in the endogenous locus but less severe embryonic lethal mutations when present in the transgene. It seems likely that differences in expression levels of the endogenous gene and transgene and/or the efficiency of RNAi might explain the differences observed. Nonetheless, the milder phenotypes associated with the transgene allowed us to make and analyze the individual S331A, S338A, and S340 mutations, whereas these could not be made in the endogenous gene.

To analyze the effects of the SAS-5$^{S10A}$ and the S331/338/340A (or SAS-5$^{3A}$) mutant, on embryonic viability, we propagated these mutants as homozygotes in the presence of ltSi357, the recoded wild-type *sas-5* transgene. As shown below, this transgene completely rescues all phenotypes associated with the S10A and 3A mutations and as mentioned earlier, because it is recoded, it can be selectively targeted for silencing by RNAi. As shown in Fig. 2C, when subjected to control (*smd-1*) RNAi, strains carrying the transgene and harboring endogenous SAS-5$^{S10A}$ or SAS-5$^{3A}$ mutations exhibited wild-type levels of embryonic viability. However, when subjected to RNAi targeting the transgene, the embryonic viability of both mutants dropped to near zero. In contrast, under the same conditions, a control strain harboring the wild-type endogenous *sas-5* gene was unaffected, demonstrating the specificity of the transgene RNAi. Thus, both the SAS-5$^{S10A}$ and SAS-5$^{3A}$ mutations completely or near completely disrupt SAS-5 function leading to failed embryogenesis. Similar results were obtained with a serine 10 to glutamate mutation (SAS-5$^{S10E}$). In summary, we have identified four conserved serine residues (S10, S331, S338, and S340) that are essential for SAS-5 function.

## Mutation of serines S331, S338, and S340 affects SAS-5 protein levels

The embryonic lethal phenotypes of the SAS-5$^{S10A}$, SAS-5$^{S10E}$, and SAS-5$^{3A}$ mutants could result from a dysfunctional SAS-5 protein or a protein that is expressed at inappropriate levels. To investigate this, we performed quantitative immunoblots on whole worm lysates. In these experiments, we used strains carrying the rescuing ltSi357 transgene and treated them with either control RNAi to determine total SAS-5

protein levels or transgene-specific (*sas-5RR*) RNAi to determine endogenous SAS-5 levels. Surprisingly, we found that when subjected to either control or *sas-5RR* RNAi, the wild-type strain carrying the *ltSi357* transgene (OD1187) expressed similar levels of SAS-5 protein (Fig. 2D,E). Two explanations could account for our failure to observe a reduction of total SAS-5 levels upon silencing of the transgene. It's possible that the transgene is expressed at a significantly lower level than the endogenous *sas-5* gene, and thus the effect of *sas-5RR* on total SAS-5 levels is minimal. Alternatively, the worm might employ a homeostatic mechanism that maintains SAS-5 protein levels within a narrow range, as has been seen for SPD-2 (Decker et al, 2011). Nonetheless, we found that upon *sas-5RR* RNAi, the levels of wild-type, SAS-5[S10A], and SAS-5[S10E] proteins were not significantly different (Fig. 2D). This indicates that the embryonic lethal phenotypes associated with the SAS-5[S10A] and SAS-5[S10E] mutants are due to a dysfunctional protein.

We next compared the levels of SAS-5[3A] and wild-type SAS-5. In the presence of *sas-5RR* RNAi, the level of SAS-5[3A] protein was found to be elevated approximately two-fold over the wild-type level (Fig. 2E); this suggests that the embryonic lethality associated with this strain is due to overexpression of SAS-5. Curiously however, as shown in Fig. 2C, the embryonic lethality of the SAS-5[3A] mutant can be rescued by expression of wild-type SAS-5 from the *ltSi357* transgene. To understand the basis for this rescue effect, we quantified the levels of wild-type SAS-5 and SAS-5[3A] under control RNAi conditions and found that the level of SAS-5[3A] is comparable to wild-type. This indicates that phosphorylation of residues 331, 338, and/or 340 negatively regulates SAS-5 levels and that this inhibitory mechanism can work in trans to control the levels of the unphosphorylated SAS-5[3A] protein.

## Serine 10 is required for centriole assembly while serines 331, 338, and 340 play a role in preventing the formation of excess centrioles

To determine the basis for the embryonic lethal phenotypes of the SAS-5[S10A], SAS-5[S10E], and SAS-5[3A] mutants, we created a set of strains that carry the aforementioned endogenous mutations and the rescuing *ltSi357* transgene; these strains also express GFP::histone H2B and GFP::γ-tubulin. Each strain was exposed to *sas-5RR* RNAi and early embryonic development recorded by multi-dimensional confocal microscopy.

Defects in centriole duplication at different developmental time-points can give rise to different patterns of mitotic spindle defects. Fig. EV3A summarizes the maternally- and paternally-controlled centriole duplication events that take place around the time of fertilization and the consequences of centriole duplication errors during each of these events. Of note, the first pair of centrioles is donated to the embryo by the sperm. Any centriole duplication defects that take place during spermatogenesis lead to sperm with abnormal numbers of centrioles and abnormal mitotic spindles during the first embryonic cell cycle (O'Connell et al, 2001; Peters et al, 2010). In contrast, centriole duplication following fertilization is under maternal control and any defects that take place during the embryonic divisions manifest as spindle defects during the ensuing cell cycle.

As expected, we found that when treated with *sas-5RR* RNAi, embryos harboring a wild-type copy of the endogenous *sas-5* gene developed normally (Fig. 3A,B). That is, such embryos invariably assembled bipolar spindles during the first cell cycle, indicating that they had inherited two sperm-derived centrioles. Also, consistent with normal centriole duplication during the first cell cycle, these embryos always assembled bipolar spindles during the second cell cycle. In contrast, when exposed to *sas-5RR* RNAi, SAS-5[S10A] and SAS-5[S10E] embryos assembled monopolar spindles during either the first and/or second cell cycles (Fig. 3C–F), indicating a defect in centriole duplication during spermatogenesis and embryogenesis respectively. In some embryos asymmetric spindles were observed; such spindles possess one pole of normal size and a second smaller pole (Fig. 3E) and have been observed under conditions where centriole assembly is partially blocked (Delattre et al, 2004; Kirkham et al, 2003). Upon quantitation of these defects, it became clear that the SAS-5[S10A] mutant has a more severe effect upon centriole duplication as almost 90% of the centrioles fail to duplicate during the first round of embryonic centriole assembly, yielding almost all monopolar spindles at the two-cell stage (Fig. 3D). In contrast in newly fertilized embryos expressing SAS-5[S10E], only 40% of centrioles failed to duplicate, resulting in mostly bipolar spindles at the two-cell stage (Fig. 3F). This indicates that SAS-5[S10E] is partially functional but not able to fully mimic wild-type SAS-5 phosphorylated at serine 10.

We next examined SAS-5[3A] embryos treated with sas-5RR RNAi and found that they looked largely normal through the first two cell cycles, as 100% of the cells assembled a bipolar spindle. However, beginning at the four-cell stage, multipolar spindles were observed in some of the cells (Fig. 3G). Approximately 30% of four-cell stage SAS-5[3A] embryos exhibited this defect and overall about 10% of cells from embryos at this stage of development underwent a multipolar division (Fig. 3H). Although not quantified, older embryos appeared to have a higher frequency of multipolar spindles. Examination of these multipolar spindles revealed that all poles contained SAS-6 (Fig. 3I), indicating the presence of centrioles. This is consistent with the extra spindle poles arising from an overduplication defect. Our results indicate that loss of phosphorylation at serine 10 leads to a block in centriole assembly while loss of phosphorylation at serines 331, 338, and/or 340 leads to centriole amplification.

To gain further insight into the effects of the various mutations on centriole assembly, we performed live imaging of embryos that expressed mutant versions of the recoded *sas-5* transgene together with GFP::histone H2B and GFP::γ-tubulin. In this genetic background, the triple mutant S10/S68/S71A corresponding to cluster I exhibited a nearly complete block in centriole duplication leading to 95% monopolar spindles at the second division (Fig. EV3B), similar to that observed for the endogenous SAS-5[S10A] mutant (Fig. 3D). However a strain expressing a transgene with just the S10A mutation alone exhibited only a fraction of two-cell embryos with monopolar spindles (Fig. EV3C), suggesting that the S68A and S71A mutations contributed to the block in centriole duplication observed in the triple mutant. To address this we analyzed centriole assembly in a strain expressing an S68/71A double-mutant transgene and surprisingly found very few two-cell embryos with monopolar spindles. Thus, while the S68/71A double mutant by itself does not strongly affect centriole assembly, it strongly enhances the phenotype of the S10A mutant. We next looked at a strain carrying a transgene expressing the equivalent of the SAS-5[3A] mutant protein and found that embryos often possessed excess centrosomes and multipolar spindles (Fig. EV3B).

   

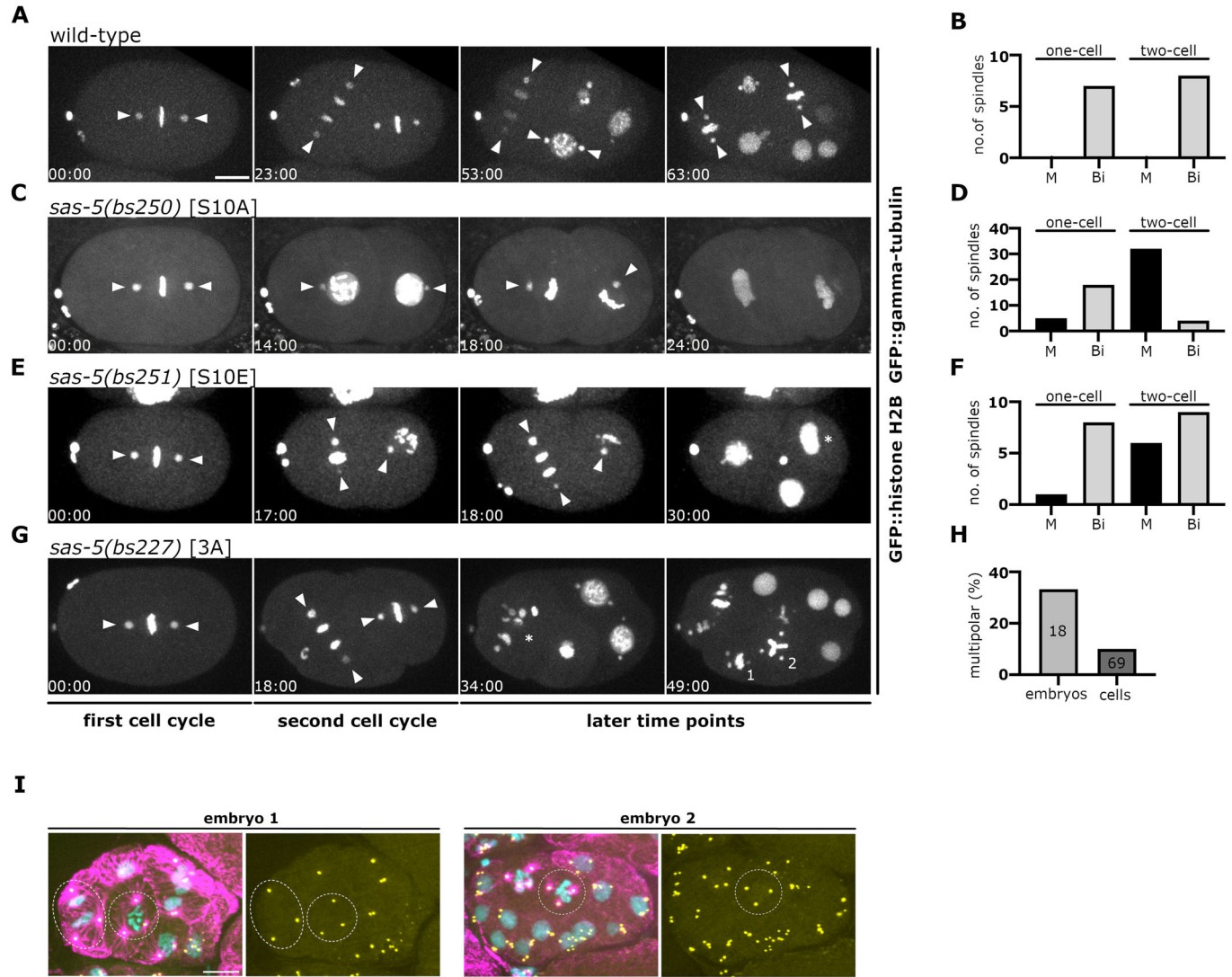

**Figure 3. The S10A mutation blocks centriole assembly, while the 3A mutation leads to multipolar spindles.**

Select frames and quantitation of spindle defects from time-lapse recordings of animals expressing (**A, B**). wild-type SAS-5, (**C, D**), SAS-5^S10A, (**E, F**) SAS-5^S10E, and (**G, H**) SAS-5^3A. The elapsed time is in minutes. All strains express GFP::Histone H2B to mark chromatin and GFP::γ-tubulin to mark centrosomes and spindle poles (arrowheads) and were fed on bacteria expressing RNAi against the *sas-5* transgene (*sas-5RR*). (**A, B**) Wild-type embryos assemble bipolar spindles during each cell cycle. (**C, D**) S10A embryos predominantly assemble monopolar spindles during the second cell cycle. (**E, F**) S10E embryos produce a mix of mono- and bipolar spindles at the two-cell stage. The embryo shown assembles a monopolar spindle in the posterior blastomere (*t* = 18:00). Following a failed nuclear division, a single nucleus (asterisk) reforms (*t* = 30:00). (**G, H**) 3A mutant embryos form multipolar spindles during later cell cycles. The 3A mutant embryo shown forms bipolar spindles during the first two cell cycles but assembles multipolar spindles at later timepoints. The multipolar spindles take two forms (*t* = 49:00): pseudobipolar (1) where two centrosomes are found on one side of a normal metaphase plate and a single centrosome on the other, and tripolar (2) where three centrosomes surround a Y-shaped metaphase plate. The asterisk (*t* = 34:00) marks a multipolar spindle dividing out of the focal plane. The scale bar in (**A**) is 10 µm and applies to (**A–D**). (**B, D, F**) The number of bipolar (Bi) and monopolar (M) spindles observed during the first and second cell cycles are shown. (**H**) The percentage of four-cell stage embryos and the percentage of cells from four-cell stage embryos with multipolar spindles are shown. The numbers inside the bars indicate the number of embryos and cells scored. (**I**) Two examples of embryos expressing SAS-5^3A stained for microtubules (magenta), endogenously tagged SPOT::SAS-6 (yellow), and DNA (cyan). Cells with multipolar spindles are circled. Note that all spindle poles contain SPOT::SAS-6 indicating the presence of centrioles. Scale bar, 10 µm. Source data are available online for this figure.

This was similar to what we found when the SAS-5^3A protein was expressed from the endogenous locus (Fig. 3G,H). Importantly however, we also examined the S331A, S338A, and S340A single mutants and found all three possessed a low frequency of multipolar spindles. Thus mutation of each of these three serines appears to contribute to the multipolar spindle phenotype observed in the SAS-5^3A mutant.

## Phosphorylation of serine 10 is required for stable incorporation of the SAS-5–SAS-6 complex into nascent procentrioles

SAS-5^S10A is a strong loss-of-function mutation that causes a near complete block in centriole assembly (Fig. 3C,D), a fully penetrant embryonic lethality (Fig. 2C), and a sterile uncoordinated

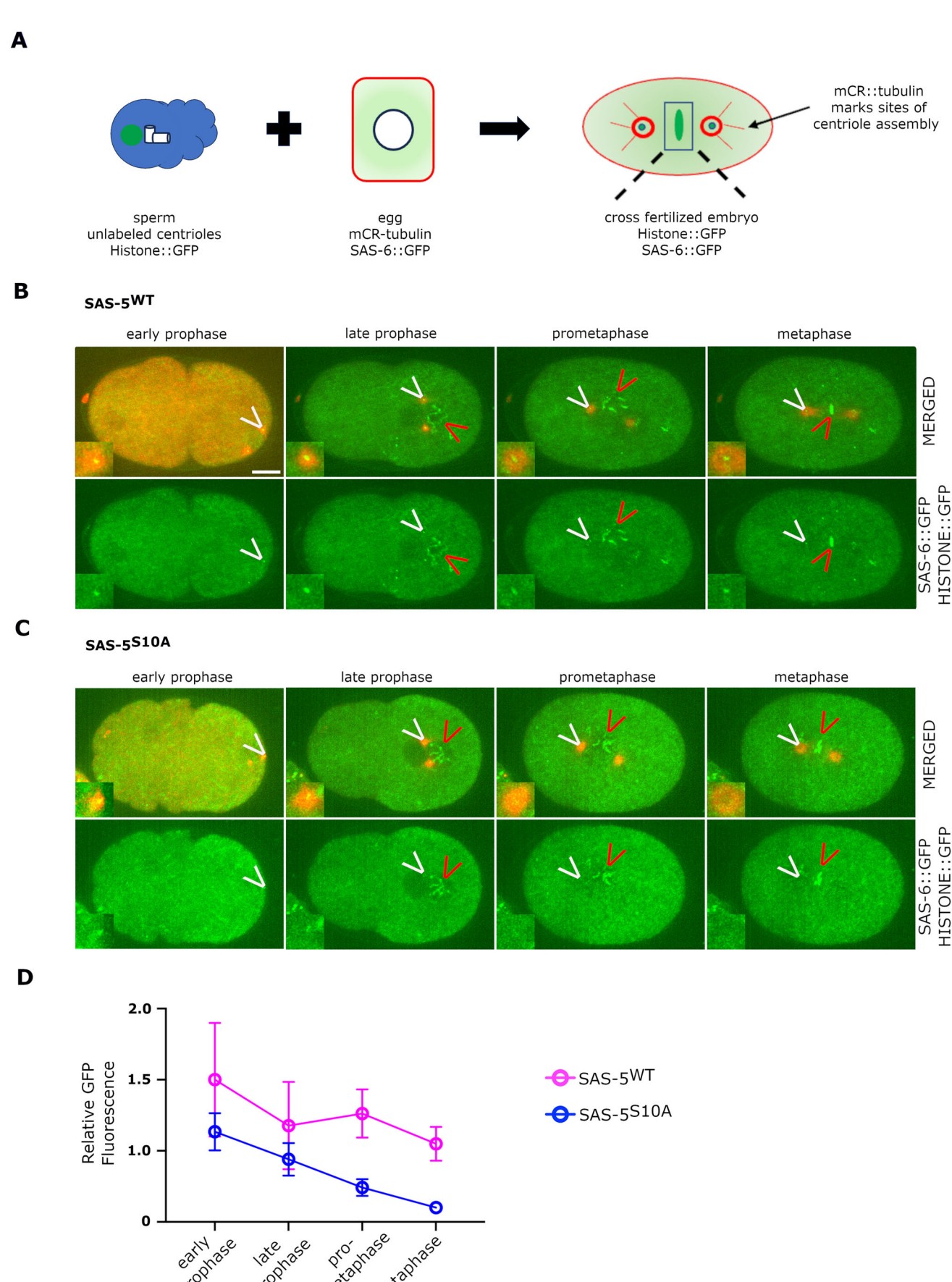

 

**Figure 4.  Phosphorylation of serine 10 is required for stable incorporation of the SAS-5–SAS-6 complex into assembling centrioles.**

(A) Schematic depicting the SAS-6::GFP recruitment assay. Males whose sperm was marked with Histone::GFP were mated to hermaphrodites expressing mCherry (mCR)::tubulin and SAS-6::GFP. Cross-fertilized embryos possess unlabeled sperm-derived centrioles and are distinguished by the presence of Histone::GFP-labeled paternal DNA. SAS-6::GFP fluorescence is then monitored within the ring of mCR::tubulin. As the sperm-derived centrioles are not fluorescently labeled, any GFP fluorescence detected at the center of the microtubule asters represents newly recruited SAS-6::GFP from the embryonic cytoplasm. (B, C) Images from a time-lapse recording of an embryo expressing SAS-5$^{WT}$ (B) and an embryo expressing SAS-5$^{S10A}$ (C). The top row shows merged images of GFP and mCherry channels, and the bottom panel shows GFP channel only. White arrowheads point to centrosomes, while red arrowheads indicate paternal DNA. Insets are approximately 3X magnifications of one of the centrosomes. Scale bar = 10 μm and applies to panels (B, C). (D) Quantification of relative SAS-6::GFP levels at centrosomes during the first round of centriole assembly. Each data point shows the mean and SEM. $n = 3$ wild-type embryos and 13 SAS-5$^{S10A}$ embryos. Note that SAS-6::GFP persists at centrioles in the wild-type embryos while it initially accumulates in the SAS-5$^{S10A}$ embryos, but is not stably incorporated and is lost. Source data are available online for this figure.

phenotype. This indicates that phosphorylation of serine 10 is a critical event in centriole duplication. We therefore set out to determine how the sas-5$^{S10A}$ mutation affects centriole assembly and analyzed the dynamic behavior of SAS-5 during the process of centriole assembly using live imaging, combined with a new method for mounting embryos (see "Methods").

SAS-5 and SAS-6 physically interact and are codependent for their recruitment to nascent procentrioles (Dammermann et al, 2004; Delattre et al, 2004; Leidel et al, 2005). Hence, either one of them can be followed to assess the behavior of the SAS-5-SAS-6 complex as a whole (Lettman et al, 2013). We therefore utilized SAS-6::GFP as a reporter and carried out a recruitment assay similar to that described previously (Lettman et al, 2013). In our version of the assay (Figs. 4A and EV4) we followed the behavior of SAS-6::GFP in zygotes produced by a cross between hermaphrodites expressing SAS-5$^{S10A}$, SAS-6::GFP, and mCherry::tubulin and males expressing histone::GFP. As zygotic centrioles are paternally inherited, the use of males lacking the SAS-6::GFP marker allowed us to follow only the newly recruited SAS-6::GFP from the maternal cytoplasm. The use of histone::GFP expressing males allowed us to unambiguously identify outcross progeny (as self-progeny lack GFP-labeled chromatin), while the use of mCherry::tubulin allowed us to mark the position of centrioles. We recorded embryos shortly after they exited meiosis. By this stage, the centriole pair associated with sperm nucleus had separated. At this stage, GFP fluorescence intensity within the tubulin ring was similar in magnitude between control and SAS-5$^{S10A}$ embryos (Fig. 4B–D). This indicates that phosphorylation of serine 10 is not required for the initial stages of recruitment. Further, this suggests that phosphorylation of serine 10 is neither required for SAS-5-SAS-6 complex formation nor for ZYG-1-SAS-6 interactions, as both are required for the initial recruitment (Gonczy, 2012; Lettman et al, 2013; Pelletier et al, 2006). As the cell cycle progressed, SAS-6 levels were found to gradually decrease in the control embryos (Fig. 4B,D), consistent with observations made in a previous study (Lettman et al, 2013). However, in the SAS-5$^{S10A}$ mutant, SAS-6::GFP was found to be rapidly lost, reaching background levels by metaphase (Fig. 4C,D). Thus, in the SAS-5$^{S10A}$ mutant, the SAS-5–SAS-6 complex is recruited, but it is not stably incorporated into the assembling centriole. Lettman et. al reported a remarkably similar set of events in embryos expressing a kinase-dead version of ZYG-1 (Lettman et al, 2013). In such embryos, the SAS-5–SAS-6 complex could initially accumulate at nascent centrioles (although to a lesser extent than wild-type embryos) before dispersing by metaphase. Taken together, these results indicate that phosphorylation of serine 10 is a prerequisite for establishing one or more molecular interactions that stabilize the centriole scaffold.

It is worth noting that the behavior of SAS-6::GFP in the presence of SAS-5$^{S10A}$ argues against the possibility that the S10A mutation leads to the production of a folding-deficient SAS-5 protein, thereby causing a complete loss of function. If it were the case, it would have also prevented its interaction with SAS-6 and SAS-6::GFP would not accumulate at the procentriole. On the contrary, we find that SAS-6 is indeed recruited to the procentrioles. Thus, the behavior of SAS-5$^{S10A}$ reflects loss of a specific function.

We next analyzed the recruitment of the downstream factor SAS-4, which is required for the stabilization of nascent procentrioles (Gonczy, 2012; Pelletier et al, 2006). SAS-4 localizes to centrioles concomitantly with the SAS-5–SAS-6 complex during first S phase (Dammermann et al, 2008) and its centriole localization depends on the presence of SAS-5–SAS-6 complex at the growing procentriole (Gonczy, 2012; Pelletier et al, 2006). To follow SAS-4, we utilized a recruitment assay very similar to the one we used for SAS-6 (Fig. 5A), except that instead of a transgene, we used endogenously tagged SAS-4::GFP.

It has previously been reported that a substantial fraction of GFP::SAS-4 expressed from a randomly integrated transgene resides in the PCM (Dammermann et al, 2008). This makes it difficult to distinguish the centriole signal from the PCM signal. However, using endogenously tagged strain, we did not observe SAS-4::GFP localized to the PCM. While we are unsure of the reason for this discrepancy, we were able to unambiguously measure SAS-4::GFP levels within an ROI of 0.7 μm in diameter. We found that in wild-type embryos, SAS-4::GFP localizes at the earliest timepoints and stays more or less steady through early mitosis (Fig. 5B,D). However, in embryos expressing SAS-5$^{S10A}$, SAS-4::GFP fails to localize to the nascent centriole (Fig. 5C,D). Thus, despite the fact that the SAS-5–SAS-6 complex localizes at least transiently to assembling centrioles in the SAS-5$^{S10A}$ mutant, SAS-4 does not become enriched at this site. This indicates that the SAS-5$^{S10A}$ mutation interferes, perhaps in a direct manner, with the ability of the SAS-5–SAS-6 complex to recruit SAS-4.

# Discussion

The *C. elegans* embryo has served as an important model system for understanding the molecular mechanism underlying centriole assembly. Indeed, many of the core components of the assembly pathway were first characterized in worms (Dammermann et al, 2004; Delattre et al, 2004; Kirkham et al, 2003; Leidel et al, 2005; O'Connell et al, 2001) with orthologs subsequently being identified in flies and humans (Basto et al, 2006; Bettencourt-Dias et al, 2005;

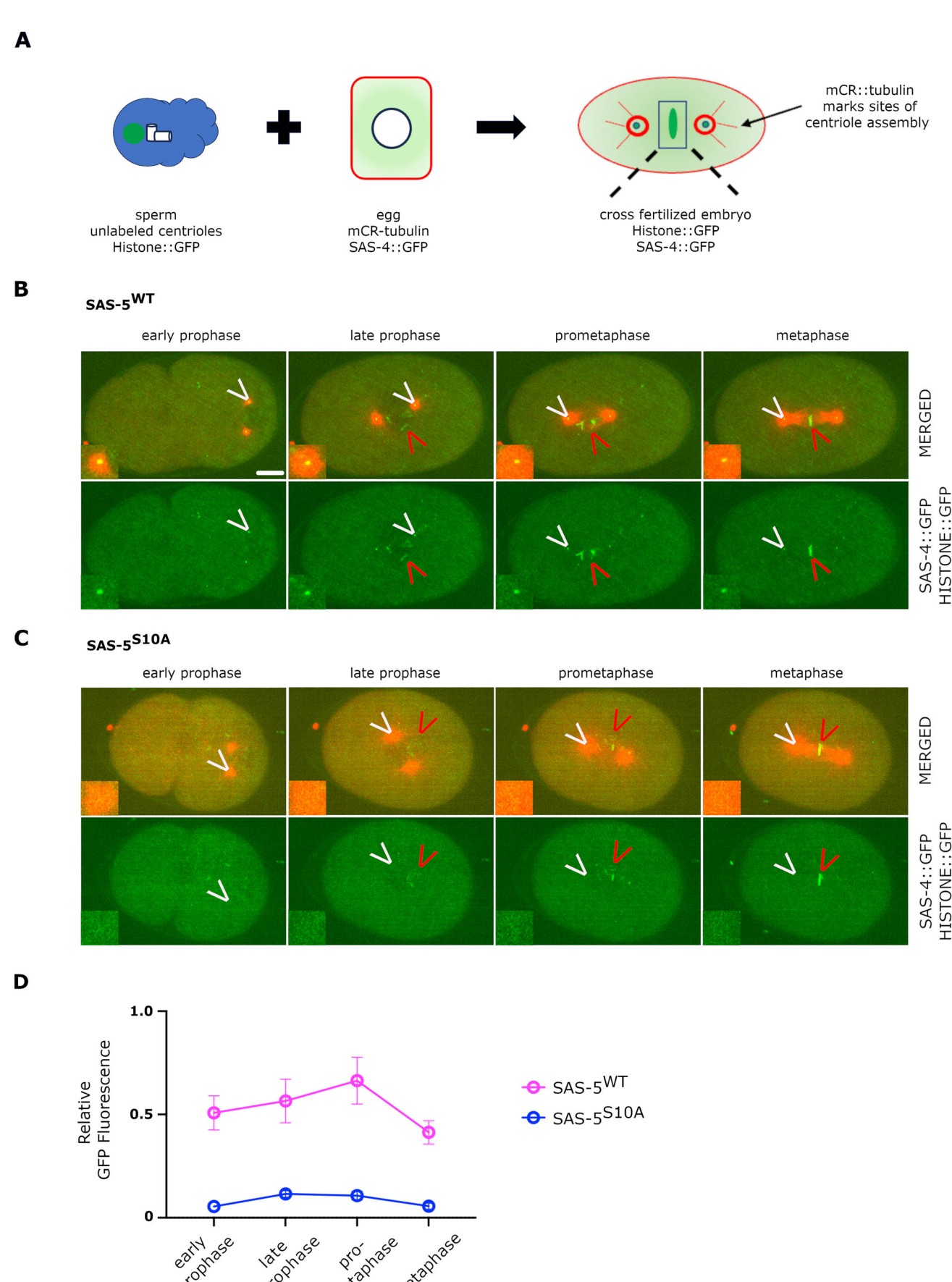

    

Figure 5.  SAS-4 is not recruited to the nascent centriole in embryos expressing SAS-5$^{S10A}$.

 (A) Schematic depicting the SAS-4::GFP recruitment assay. Males whose sperm was marked with Histone::GFP were mated to hermaphrodites expressing mCR-tubulin and endogenously tagged SAS-4::GFP. Cross-fertilized embryos possess unlabeled sperm-derived centrioles and are distinguished by the presence of Histone::GFP-labeled paternal DNA. SAS-4::GFP fluorescence is then monitored within the ring of mCR-tubulin. As the sperm-derived centrioles are not fluorescently labeled, any GFP fluorescence detected at the center of the microtubule asters represents newly recruited SAS-4::GFP from the embryonic cytoplasm. (B, C) Images from a time-lapse recording of an embryo expressing SAS-5$^{WT}$ (B) and an embryo expressing SAS-5$^{S10A}$ (C). The top row shows merged images of GFP and mCherry channels, and the bottom panel shows the GFP channel only. White arrowheads point to centrosomes, while red arrowheads indicate paternal DNA. Insets are approximately 3X magnifications of one of the centrosomes. Scale bar = 10 μm and applies to (B, C). (D) Quantification of relative SAS-4::GFP levels at centrosomes during the first round of centriole assembly. Each data point shows the mean and SEM. $n = 7$ for both wild type and SAS-5$^{S10A}$. Note that SAS-4::GFP accumulates at centrosomes at the earliest stages in the wild-type embryos but fails to do so in SAS-5$^{S10A}$ embryos. Source data are available online for this figure.

Kleylein-Sohn et al, 2007; Leidel et al, 2005; Rodrigues-Martins et al, 2008; Stevens et al, 2010). Work in all three systems has led to the identification of a core conserved centriole assembly pathway where the master regulator ZYG-1/Plk4 recruits two components of the centriole scaffold SAS-6 and SAS-5/Ana2/STIL which in turn recruit the microtubule-binding component SAS-4/CPAP (Delattre et al, 2006; Pelletier et al, 2006). Work in human cells and flies has established Ana2/STIL as one of the primary substrates of PLK4, and that phosphorylation at the C-terminus of Ana2/STIL promotes its association with SAS-6 (Dzhindzhev et al, 2014; Ohta et al, 2014) while phosphorylation at a separate region of Ana2/STIL promotes its interaction with the TCP domain of CPAP (McLamarrah et al, 2020; Moyer and Holland, 2019). There is no evidence in worms that phosphorylation is needed for the SAS-5–SAS-6 interaction. However, Cottee et al have shown that the TCP domain of SAS-4 is required for its centriolar localization and centriole duplication in *C. elegans* (Cottee et al, 2013), suggesting that the SAS-4–SAS-5 interaction is conserved in worms and could be a target of ZYG-1-mediated phosphorylation.

To date, nearly all of the in vitro work on centriole assembly has utilized protein fragments, as full-length proteins are difficult to express and purify. While such studies have proven useful for dissecting some of the protein interactions and phosphorylation events that take place during centriole assembly, it may be that protein fragments do not always faithfully mimic the behavior of full-length proteins. With this in mind, we devised a strategy to express and purify full-length ZYG-1, SAS-4, SAS-5, and SAS-6 from *E. coli*. We found that by using a rapid induction protocol, we could drive high-level expression of these proteins in inclusion bodies, which could be purified to near homogeneity and refolded in vitro. Each of the refolded proteins exhibited a CD spectrum consistent with the secondary structural elements present in the known or predicted structure of the protein. Furthermore, ZYG-1 regained kinase activity, was able to phosphorylate proteins at a nanomolar concentration, and showed a clear substate preference for SAS-5 (Fig. 1C).

Utilizing these full-length proteins, we were able to show that relative to SAS-4 and SAS-6, SAS-5 is strongly phosphorylated by ZYG-1 in vitro (Fig. 1C). We find that full-length SAS-5 can interact with multiple individual ZYG-1 domains including the kinase domain, cryptic polo-box (CPB) and PB3 domains (Fig. 1H,I). Similarly in both humans and flies, multiple Plk4 domains, including the kinase domain, the L1 linker, and the polo-box domains, have been found to interact with STIL/Ana2 (Arquint et al, 2015; Cottee et al, 2017; Kratz et al, 2015; McLamarrah et al, 2018; Ohta et al, 2014). However, the exact domains identified as interactors, varied from one study to the next making it difficult to

draw a consensus. Nonetheless, it is likely that in worms, flies and humans, multiple domains of ZYG-1/Plk4 contribute to binding SAS-5/Ana2/STIL.

SAS-5 is a rapidly diverging protein with some distantly related nematode orthologs sharing as little as 25% identity with *C. elegans* SAS-5. We find it remarkable that almost half of the 37 serine and threonine residues phosphorylated by ZYG-1 in vitro are conserved among divergent nematodes. This suggests that the residues phosphorylated by ZYG-1 are more likely to be under selective pressure to be maintained. Indeed, mutation of at least seven of the conserved serine and threonine residues (serines 10, 301, 304, 331, 338, and 340 and threonine 101) led to a moderate to strong phenotype. Together our observations suggest that the residues phosphorylated by ZYG-1 in vitro are important for SAS-5 function in vivo.

Of particular interest to us are residues whose mutation to alanine results in animals that are sterile and uncoordinated, a phenotype observed in our presumptive null allele *sas-5(bs267)*. Just two mutations produce such a phenotype: at the N-terminus, mutation of serine 10 to alanine and at the C-terminus, mutation of serines 331, 338, and 340 to alanine. Serine 10 appears to be the only serine or threonine residue in SAS-5 whose phosphorylation by ZYG-1 is indispensable for centriole assembly. In the absence of serine 10 phosphorylation, the SAS-5–SAS-6 complex can accumulate at the nascent centriole but is not stably incorporated and diffuses away before mitosis. This is similar to the behavior of the SAS-5–SAS-6 complex in embryos expressing a kinase-dead version of ZYG-1 (Lettman et al, 2013) or when SAS-4 is knocked down (Pelletier et al, 2006). Overall, our results are consistent with a model whereby ZYG-1 executes two distinct steps in the centriole assembly pathway (Fig. 6). First, ZYG-1 functions in a kinase-independent step to recruit the SAS-5–SAS-6 complex via direct physical interactions with both SAS-5 and SAS-6. Second in a kinase-dependent step, ZYG-1 phosphorylates serine 10 of SAS-5 to drive stable cartwheel assembly.

Our results show that the S10A mutation also blocks the recruitment of SAS-4. Since SAS-4 is also required for stable association of the SAS-5–SAS-6 complex with the nascent centriole (Pelletier et al, 2006), it is tempting to speculate that serine 10 phosphorylation promotes SAS-5–SAS-4 interactions, similar to what has been seen in vertebrates and flies (McLamarrah et al, 2020; Moyer and Holland, 2019). In *Drosophila*, humans and in *Danio rerio* the STIL/CPAP interaction is mediated through a conserved proline-rich motif in STIL and the G-box/TCP domain of CPAP (Cottee et al, 2013; Hatzopoulos et al, 2013; Tang et al, 2011). However, such a proline-rich motif can not be unambiguously identified in *C. elegans* SAS-5 as well as other nematodes

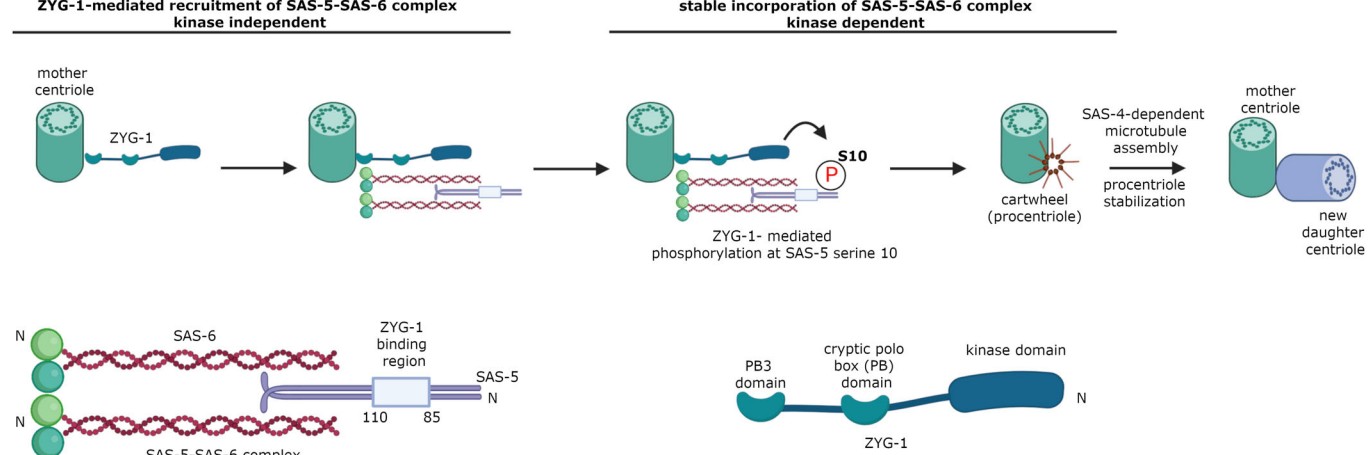

**Figure 6. A model depicting the actions of ZYG-1 during centriole assembly.**

In a kinase-independent step, ZYG-1 tethered to the mother centriole wall recruits the SAS-5–SAS-6 complex from the cytoplasm. This step depends on a direct physical interaction with SAS-6. ZYG-1 also physically interacts with SAS-5, but whether this interaction is required for recruitment of the complex is not presently known. In a subsequent kinase-dependent step, ZYG-1 phosphorylates SAS-5 on serine 10 to drive stable incorporation of the SAS-5–SAS-6 complex and cartwheel formation. SAS-4 which is recruited to the nascent centriole concomitantly with the SAS-5–SAS-6 complex is required to complete centriole assembly via the addition of microtubules to the outer wall. When serine 10 phosphorylation is blocked, SAS-4 is not recruited to the nascent centriole suggesting that the failure to stably incorporate the SAS-5–SAS-6 complex is due to a weakening of the SAS-4–SAS-5 interaction. Model produced using BioRender.

(Cottee et al, 2013). Thus, it may be that the interaction interface of the SAS-4–SAS-5 complex, like that of the ZYG-1–SAS-5 complex differs between vertebrates and worms. Future studies will be required to probe the SAS-4–SAS-5 interaction and determine how it might be regulated by ZYG-1.

In vitro, ZYG-1 also phosphorylates SAS-5 on residues S331, S338, and S340 at the C-terminus. Mutation of all three sites to alanine results in a recessive sterile uncoordinated phenotype, suggestive of a complete loss of function. However, our analysis revealed that the SAS-5[3A] protein is overexpressed and results in the formation of multipolar spindles, consistent with a gain-of-function effect that drives the formation of extra centrioles. The fact that this mutation is not dominant and can be rescued by a wild-type transgene suggests that the wild-type SAS-5 protein can act in trans to downregulate the SAS-5[3A] mutant protein and likely reflects the fact that SAS-5 exists in an oligomeric state (Lettman et al, 2013; Qiao et al, 2012). Consistent with this notion, quantitative immunoblotting revealed that the SAS-5[3A] protein is expressed at wild-type levels in the presence of the wild-type transgene but become elevated only upon transgene silencing. While further study will be required to understand the underlying mechanism, the role of ZYG-1 in this process is presently unclear. Depletion of ZYG-1 does not result in elevation of SAS-5 levels. Thus, one possibility is that some other kinase by itself or in combination with ZYG-1 phosphorylates these SAS-5 sites in vivo. Alternatively, it could be that ZYG-1 is the one and only kinase, but that ZYG-1 normally only functions locally to control the level of SAS-5 at the centriole. Interestingly, serines S331, S338, and S340 lie within the LET-92-binding site of SAS-5 (Kitagawa et al, 2011a). We have earlier shown that knockdown of LET-92, the sole catalytic subunit of the *C. elegans* Protein Phosphate 2A (PP2A) complex, leads to a decrease in SAS-5 levels (Song et al, 2011). This suggests that serines S331, S338, and S340 are the targets of PP2A, and that

kinase and phosphatase coordinately operate through these residues to control the level of SAS-5.

Previous work has established that ZYG-1 binds directly to SAS-6 and that this interaction is required for the recruitment of the SAS-5–SAS-6 complex (Lettman et al, 2013). Here we show that ZYG-1 also physically associates with SAS-5 (Fig. 1D–F). In light of this, it is interesting to note that ZYG-1 may have evolved a distinct mechanism for docking onto SAS-5. In flies and humans, Plk4 binds to the coiled-coil domain of STIL/Ana2 (Arquint et al, 2015; Cottee et al, 2017; McLamarrah et al, 2018; Ohta et al, 2014). While the limitations of our interaction assay leave open the possibility that ZYG-1 also binds the coiled-coil region of SAS-5, our results indicate that ZYG-1 makes important contacts outside of this region (Fig. 1D–F). Perhaps because ZYG-1 binds directly to SAS-6 (Lettman et al, 2013) but Plk4 does not, the ZYG-1–SAS-5 interaction might be under less selective pressure than the PLK4–STIL/Ana2 interaction. That is by maintaining a strong independent interaction with SAS-6, the ZYG-1–SAS-5 interface could freely evolve over time. So is the ZYG-1–SAS-5 interaction essential or is the ZYG-1–SAS-6 interaction sufficient to recruit the SAS-5–SAS-6 complex? We attempted to make a deletion of the conserved tract of amino acids (spanning 68–110) around the ZYG-1 binding site in SAS-5, but were only able to recover sterile heterozygotes, indicating that such a mutation has a strong dominant-negative effect. Thus it remains possible that ZYG-1 needs to bind both components of the SAS-5–SAS-6 complex (Lettman et al, 2013).

While there is clearly some evolutionary divergence regarding the specific molecular interactions that take place during centriole assembly, the overall general theme appears to be conserved among nematodes, flies and humans. Nonetheless, multiple examples of mutations having tissue-specific effects in centriole assembly suggest that the basic process might be altered in different cell types. For

   

instance, C-terminal deletions of ZYG-1 behave as loss-of-function alleles in the female germ line, while exhibiting gain-of-function characteristics in the male germ line (Peters et al, 2010). Similarly, worms homozygous for the *sas*-5(*t2079*) allele, which express a SAS-6-binding deficient version of SAS-5, develop fully functional germ lines, while producing embryos in which centriole assembly is completely blocked (Delattre et al, 2004). This indicates that SAS-5–SAS-6 oligomerization is either dispensable in the germline or is achieved through a distinct molecular interface. Finally in humans, mutations in genes encoding core centriole assembly factors disrupt brain development without grossly affecting many other tissues (Phan and Holland, 2021). All of these examples underscore the need to assay mutant constructs in the context of a multicellular animal. Here we have done that and have shown that the SAS-5^S10A and SAS-5^3A mutations disrupt embryonic and postembryonic divisions, suggesting that these phosphorylation events are likely to be broadly required for centriole assembly.

Our results fill a critical gap in our knowledge of how phosphorylation drives centriole assembly in *C elegans*, an organism that has played a central role in understanding this process. We find that similar to other organisms (McLamarrah et al, 2020; Moyer and Holland, 2019), ZYG-1 phosphorylates the centriole scaffold protein SAS-5, to stably incorporate the SAS-5–SAS-6 complex and SAS-4 into the nascent centriole. In vertebrates and flies, Plk4 also phosphorylates STIL/SAS-5 within the STAN domain to promote association with SAS-6 (Dzhindzhev et al, 2014; Ohta et al, 2014). However, this does not appear to be the case in *C. elegans* where SAS-5 and SAS-6 can associate and localize to nascent centrioles in the absence of ZYG-1 kinase activity (Lettman et al, 2013). Thus, the critical role of ZYG-1 kinase activity with respect to the SAS-5–SAS-6 complex is to promote its stable association with the growing centriole.

# Methods

## Protein expression

The coding regions of *zyg-1*, *sas-5*, *sas-6*, and *sas-4* were codon optimized for expression in *E. coli* and synthesized by Genewiz inc (Burlington, MA, USA). For *sas-5* and *sas-6*, a glycine codon was introduced immediately next to initiator methionine to satisfy the "N-end rule" (Tobias et al, 1991). The genes were cloned into the pET21b (Novagen, now Millipore Sigma, St. Louis, MO, USA) vector which provides a 6X-His tag at the C-terminus. *zyg*-1 was cloned between NcoI and HindIII sites, while the others were cloned between NdeI and HindIII. The restriction enzymes were purchased from New England Biolabs (Ipswich, MA, USA). The plasmid containing *zyg*-1 was transformed into T7 Express lysY/Iq *E. coli* (New England Biolabs) to reduce the gene toxicity associated with leaky expression; the others were transformed into BL21(DE3) *E. coli* (New England Biolabs). Large-scale cultures were grown at 37 °C with shaking at 250 rpm to an $OD_{600}$ of 0.6 and induced with 1 mM IPTG. The cultures were maintained at 37 °C for another 4 h with shaking. Samples were drawn at an interval of 2 h to assess protein expression. The cultures were split into 150 mL fractions and centrifuged at $2800 \times g$ at 4 °C for 10 min. The pellet was washed with 50 mM Tris pH 8.0, containing 250 mM NaCl (buffer A), and stored at −20 °C.

## Protein isolation, purification, and in vitro refolding

Each bacterial pellet was resuspended in 8 mL cold buffer A with 5 mM DTT. PMSF (phenylmethylsulfonyl fluoride) was added to a final concentration of 1 mM. 200 μL Antifoam (Millipore Sigma) was added to the suspension before lysing cells using a Constant Systems Limited (Northants, UK) continuous flow cell disruptor at 4 °C. The lysates were cleared by centrifugation at $20{,}000 \times g$ at 4 °C for 15 min. The pellet was resuspended in buffer A and washed twice with intervening centrifugation steps as mentioned above.

The pellet containing the inclusion bodies was solubilized in cold 8 M urea in buffer A containing 5 mM DTT, and gently rotated for 30 min at 4 °C. The solubilized pellet was centrifuged as above to remove insoluble debris. The supernatant was applied onto cOmplete™ His-Tag Purification Resin (Millipore Sigma) pre-equilibrated with buffer A containing 8 M urea and 5 mM DTT. The resin-supernatant slurry was gently mixed at 4 °C for an hour. The slurry was allowed to settle, and the supernatant was drained under gravity. The column was washed with five bed volumes of 8 M urea in buffer A containing 5 mM DTT to remove unbound proteins. The proteins were eluted with buffer A containing 60 mM Imidazole, 8 M urea, and 5 mM DTT.

The eluate was then dialyzed in a Float-A-Lyzer™ G2, NMCO 20 kDa (SpectraPor, now Repligen, Waltham, MA, USA) against buffer A with 5 mM DTT at 4 °C, with rapid buffer changes. DTT was added fresh during each buffer change. Following dialysis, the refolded protein solution was centrifuged at $17{,}000 \times g$ at 4 °C to remove particulates. The refolded proteins were then subjected to a second round of dialysis against the following buffers: for ZYG-1, 50 mM Tris, pH 8.0 containing 50 mM NaCl, 2 mM TCEP; for SAS-5, 50 mM Tris, pH 8.0, 300 mM NaCl, 2 mM TCEP; for SAS-6, 50 mM Tris, pH 7.4, 150 mM NaCl; for SAS-4, buffer A with 2 mM TCEP. ZYG-1, SAS-5, and SAS-6 were passed through Superose 6 (Cytiva, Marlborough, MA, USA) gel filtration column pre-equilibrated with the respective buffers to remove aggregates. The fractions containing proteins were pooled and concentrated.

Circular Dichroism (CD) spectra for ZYG-1, SAS-5, and SAS-6 and SAS-4 was recorded on a Jasco 815J spectropolarimeter (Jasco, Easton, MD, USA) with a protein concentration of 0.2 mg/ml, at 25 °C. The data was recorded at a scan speed of 100 nm/sec, with an integration time of 1 s. The represented spectra are the average of four consecutive accumulations.

## In vitro kinase assays

Kinase assays contained 60 nM wild-type or kinase-dead ZYG-1::6XHis and either SAS-6::6XHis, SAS-5::6XHis or GST::SAS-4::6XHis at concentrations of 1, 2, and 3 μM in 20 mM Tris pH 7.5, 50 mM NaCl, 10 mM $MgCl_2$, 1× PhosSTOP pan phosphatase inhibitor (Millipore Sigma), 1 mM DTT, 0.2 mM ATP, and 0.2 μM (γ^32P)-ATP (3000 Ci/mmol) (Perkin Elmer, Shelton, CT, USA), and were incubated for 30 min at 30 °C. The reactions were stopped by the addition of SDS-PAGE sample buffer and separated on a 4–15% Mini-PROTEAN® TGX™ Precast Protein Gel (Bio-Rad, Hercules, CA, USA). Bands were visualized by exposing gels to a BAS-IP MS Storage Phosphor Screen (GE Healthcare, Marlborough, MA, USA) for 45 min at room temperature. The screens were scanned on a FLA5100 Phosphorimager (FujiFilm, USA).

## Protein purification for pull-down assays

All constructs used in GST pull-down experiments contained the specified ORF with a C-terminal 6X-His tag cloned downstream of the GST sequence in the pDEST15 vector (Thermo Fisher Scientific, Grand Island, NY USA). N-terminal truncations, amino acid substitutions (Ser -> Ala and Ser -> Glu) and deletions in *sas-5* were introduced using Q5® Site-Directed Mutagenesis Kit (New England Biolabs). All the GST variants except the ZYG-1 fragment comprising the cryptic polo box (CPB) spanning 338–564, were purified and refolded in vitro as mentioned above. The ZYG-1 CPB was purified as reported by Shimanovskaya et al (Shimanovskaya and Dong, 2014).

## Immunoblotting

Quantitative immunoblotting was performed as previously described (Iyer et al, 2022). Briefly, adult worms were rinsed off plates in M9 buffer (22 mM $KH_2PO_4$, 22 mM $Na_2HPO_4$, 85 mM NaCl, 1 mM $MgSO_4$), washed five times in M9 buffer, suspended in homemade 4X LDS-NuPAGE sample buffer with Orange G as tracking dye to a final concentration of 1X, and sonicated using a Branson Sonicator fitted with a microtip probe. Five pulses of a 50% duty cycle with output control set at 6 was used to lyse worms and shear DNA. For each sample, proteins from the equivalent of 50 gravid adults were fractionated on a NuPAGE 4–12% Bis–Tris precast gel (Thermo Fisher Scientific). The gel was then blotted to a nitrocellulose membrane using the iBlot semi-dry transfer system (Thermo Fisher Scientific) according to the manufacturer's instructions. The membranes were blocked in Odyssey blocking buffer (LiCOR Biosciences, Lincoln, NE, USA) and probed with 1:1000 dilutions of rabbit anti-SAS-5 (Iyer et al, 2022) and DM1A, a mouse monoclonal anti-alpha-tubulin antibody (Millipore Sigma) as primary antibodies. The membrane was washed 5× in PBST buffer and probed with anti-mouse 680 and anti-Rabbit 800 IRDye secondary antibodies (LiCOR Biosciences) at a 1:14,000 dilution for 1 h at room temperature. After washing five more times as before, membranes were imaged using the Odyssey Clx imaging system (LiCOR Biosciences) and band intensities quantitated using Fiji software.

## Pull-down assays

For GST pull-down assays, 20 µL of a suspended slurry of magnetic glutathione beads (ThermoFisher Scientific) was aliquoted into tubes. The beads were washed and equilibrated with buffer A containing 5 mM DTT. 4.32 mM of bait proteins were incubated with the beads for 1 h at 4 °C on a rotator. The beads were then washed five times with the same buffer. The beads were blocked with Odyssey® Blocking Buffer (Li-Cor Biosciences) for 1 h at 4 °C on a rotator and washed three times to remove excess blocking buffer. Following blocking, 0.2 mM prey proteins were incubated with the beads for an hour at 4 °C on a rotator. Beads with GST alone served as negative controls. When ZYG-1 was used as prey, the beads were washed with 50 mM Tris, pH 8.0 buffer containing 50 mM NaCl, 5 mM DTT. For SAS-5 as prey, a solution of 50 mM Tris, pH 8.0, 300 mM NaCl, 5 mM DTT was used as wash buffer. Washed beads were then resuspended in LDS-NuPAGE sample

buffer and analyzed by immunoblotting as described above with 1:1000 dilutions of mouse monoclonal anti-GST antibody (Santa Cruz Biotechnology, Dallas, TX, USA) and rabbit anti-ZYG-1 antibody (O'Connell et al, 2001) or rabbit anti-SAS-5 (Iyer et al, 2022) as primary antibodies.

## Mapping SAS-5 phosphorylated residues

For identifying phosphorylated residues in SAS-5 by mass spectrometry, 60 nM ZYG-1 was incubated with 2 mM SAS-5 in 20 mM Tris pH 7.5, 50 mM NaCl, 10 mM $MgCl_2$, 1× PhosSTOP (Millipore Sigma), 1 mM DTT, 1 mM ATP at 30 °C for 30 min. The reaction was stopped by addition of 20 µL glacial acetic acid. Mass spectrometry was performed at the Whitehead Proteomics Facility, Cambridge, MA, USA. Samples were resuspended in 8 M Urea 50 mM Tris pH 8.0, reduced with 10 mM TCEP for 30 min and alkylated with 5 mM fresh iodoacetamide for 30 min in the dark. Samples were digested overnight in the presence of 1 mM $CaCl_2$ and trypsin at a 1:50 enzyme-to-substrate ratio. Digested samples were acidified to 5% final formic acid and centrifuged for 30 min. Phosphopeptides were enriched using Titanium dioxide matrix (Thingholm and Larsen, 2016). The enriched samples were loaded into a Thermo Orbitrap Elite with a Waters NanoAcuity UPLC with ESI (nanospray). The initial MS scan was performed with FT-ICR/Orbitrap. The MS/MS Scan mode was set to Linear Ion Trap. The resulting peptide spectrum data was searched using PEAKS algorithm against a custom-made database containing *C. elegans* sequences. A False Discovery Rate (FDR) was set to 0.9%. The validity of peptide/spectrum matches was assessed using a defined parameter—"Ascore", which calculates an ambiguity score as $-10 \times log10$ P. The *P* value indicates the likelihood that the peptide is matched by chance (Zhang et al, 2012). Peptides scoring above 18 were considered for in vivo mutational scanning.

## Worm strains and maintenance

Worms were grown at 16–25 °C on MYOB plates seeded with *E. coli* OP50 and maintained according to standard protocols (Brenner, 1974) All worm strains used in this study are listed in Dataset EV2.

## CRISPR/Cas9-mediated genome editing

Worms were injected with in vitro preformed ribonucleoprotein complexes containing Cas9, tracrRNA, and crRNA as described (Paix et al, 2015). For screening, we used a modified coCRISPR strategy that employed plasmids expressing red fluorescent proteins (RFP) as co-injection markers. These plasmids, pCFJ104 (*Pmyo-3::mCherry::unc-54utr*) and pCFJ90 (*Pmyo-2::mCherry::unc-54utr*) were injected at concentrations of 5 and 2.5 ng/ml, respectively (Frokjaer-Jensen et al, 2008). $F_1$ offspring of injected animals expressing the co-injection markers were screened for the desired *sas-5* mutation by restriction fragment length polymorphism analysis. Independent edited lines were sequenced to confirm the mutation and backcrossed twice to wild-type worms before analysis. The sequences of gRNAs and repair templates are listed in Dataset EV3. The gRNAs were designed using the CRISPOR online tool (Concordet and Haeussler, 2018).

    

## MosSCI transgenesis

A transposon-based strategy (Frokjaer-Jensen et al, 2008) was used to generate all *sas-5* transgenes. The sequence encoding the first 157 amino acids of the transgenes was recoded by codon shuffling to allow selective RNAi against endogenous or transgenic *sas-5*. These transgenes were cloned into the pCFJ151 backbone for targeted insertion at the ttTi5605 locus on chromosome II. Single-copy insertion transgenes were generated by injecting a mixture containing the transgene-encoding plasmid (50 ng/µl), the transposase plasmid (pCFJ601/Peft-3::Mos1 transposase, 50 ng/µl), three fluorescent negative selection markers (pCFJ90/Pmyo-2::mCherry, 2.5 ng/µl, pCFJ104/ Pmyo-3::mCherry, 5 ng/µl, and pGH8/Prab-3::mCherry, 10 ng/µl), and an additional negative selection marker (pMA122/Phsp-16.41::peel-1, 10 ng/µl) into the gonads of EG6699 (ttTi5605) worms. After 1 week, progeny of injected worms were heat shocked at 34 °C for 2–4 h to induce PEEL-1 expression to kill worms harboring extra chromosomal arrays (Seidel et al, 2011). Motile worms without fluorescent markers were identified and transgene integration was confirmed in their progeny by PCR spanning both homology regions.

## RNA-mediated interference (RNAi) and quantification of embryonic lethality

For RNAi against the *sas-5* transgene, the recoded region (the first 630 bp) was cloned to the dsRNA Gateway expression vector, pCR88 (Golden and O'Connell, 2007) and transformed into HT115(DE3) *E. coli*. dsRNA was introduced by feeding as described previously (Kamath et al, 2003). Briefly, L1 (SAS-5[S10A] and SAS-5[S10]) or L3 (SAS-5[3A]) larvae were placed onto a lawn of dsRNA-expressing bacteria that had been grown on MYOB plates supplemented with 50–100 µg/ml carbenicillin, with 25 µg/ml tetracycline and 1–2 mM IPTG (Isopropyl β-d-1-thiogalactopyranoside). Worms were left on plates for 24–36 h (SAS-5[3A]) or transferred after 24 h to fresh RNAi plates (SAS-5[S10A] and SAS-5[S10E]) and incubated for another 24–36 h. Embryonic lethality was then quantified over the next 24 h. RNAi against the nonessential *smd-1* gene served as a negative control.

To quantify embryonic lethality in CRISPR-derived strains that were homozygous viable, L4 larvae were singled to individual 35 mm MYOB plates and incubated at 16, 20, or 25 °C for 48, 24, or 12 h, respectively. The parent was removed, and the plates were incubated overnight before larvae and unhatched embryos were counted.

To silence the endogenous *sas-5* gene in strains containing wild-type and mutant versions of the *sas-5* transgene, 1 mg/ml of dsRNA corresponding to the sequence encoding the first 157 amino acids of endogenous SAS-5 was injected into both gonads of adult hermaphrodites (1 day after the L4 stage). To measure embryonic lethality, injected worms were singled out onto individual plates 24 h post injection and removed from the plates 24 h later. Embryonic lethality was quantified by counting the number of hatched vs. unhatched worms one day after the mother was removed. For imaging of embryos after endogenous *sas-5* knockdown, embryos were obtained by dissection between 24 h and 32 h post injection.

## Confocal imaging of live and fixed specimens

Immunostaining of embryos was performed essentially as described (O'Connell and Golden, 2014). DM1A and rabbit anti-GIP-1 antibody (Hannak et al, 2002) were used at dilutions of 1:500 and 1:1000, respectively. Alexa 568 anti-mouse and Alexa 488 anti-rabbit secondary antibodies (Thermo Fisher Scientific) were used at a 1:1000 dilution. Chromotek (Rosemont, IL, USA) SPOT-label Alexa 488 was used at 1:1000.

For spinning disk confocal microscopy, we used a Nikon Eclipse Ti2 microscope equipped with a Plan Apo 60 × 1.4 N.A. oil immersion lens, a CSU-X1 confocal scanning unit (Yokogawa Electric Corporation, Tokyo, Japan), and an ORCA-fusion BT CMOS camera (Hamamatsu, Shizuoka Pref., Japan). Excitation light was generated using 405 nm, 488 nm, and 561 nm solid-state lasers housed in an LU-NV laser unit. NIS-Elements software (Nikon Instruments, Inc, Tokyo, Japan) was used for image acquisition and initial processing.

For live imaging, a precut, 1-mm thick CultureWell™ Reusable Silicone Gasket (Grace Biolabs, Oregon, USA) with a circular well of 1.5-mm diameter was placed on a glass slide that had been precleaned of lint using cellophane adhesive tape. A drop of molten 3% agarose in Egg Buffer (118 mM NaCl, 48 mM KCl, 2 mM CaCl₂, 2 mM MgCl₂, 25 mM HEPES, pH 7.3) was placed in the circular well and another precleaned slide was placed on top to cast an agar pad. Worms were dissected in 0.5 µL Egg buffer on an 18 mm × 18 mm no. 1.5 coverslip. The agar pad was inverted on the coverslip and gently pressed to form an airtight seal between the gasket and the coverslip and imaged at 20 °C as above.

## Centriole protein recruitment assays

The strain used in the SAS-6::GFP recruitment assay was constructed as described in Fig. EV4. The mating between the parental strains was performed on MYOB plates with bacteria expressing dsRNA against the recoded region of the *sas-5* transgene. The resultant F1 progeny at L1/L2 larval stage that lacked the *nT1* balancer were picked to a fresh lawn of bacteria expressing the dsRNA and allowed to grow for 24 h at 20 °C. Subsequently, the worms were transferred to a new bacterial lawn expressing the dsRNA and allowed to mate with males expressing histone::GFP for 12 h at 20 °C. The worms were dissected, and live embryos with histone::GFP were imaged as described above.

For the SAS-4 recruitment assay, strains expressing the recoded SAS-5 transgene, SAS-4::GFP, mCherry::tbb-2, and SAS-5[S10A] or SAS-5[WT] were used. Unexpectedly, the SAS-5[S10A] strain exhibited a fully penetrant embryonic lethal phenotype when it harbored two copies of SAS-4::GFP, and had to be maintained as a balanced heterozygote using the balancer *hT2* (OC1240). SAS-4::GFP in such a strain was highly prone to bleaching even at low laser power, hence worms lacking the balancer were used for the assay. L1 stage larvae were picked onto a bacterial lawn expressing dsRNA against the transgene and processed as described for the SAS-6::GFP recruitment assay.

# Data availability

This study includes no data deposited in external repositories.

The source data of this paper are collected in the following database record: biostudies:S-SCDT-10_1038-S44319-024-00157-y.

# Peer review information

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

## Acknowledgements

The authors thank Yao Liang Wong for strain construction. Research within the KFOC lab was supported by the Intramural Research Program of the National Institutes of Health and the National Institute of Diabetes and Digestive and Kidney Diseases. KFO was supported by a National Institutes of Health grant GM074207.

## Author contributions

**Prabhu Sankaralingam**: Conceptualization; Data curation; Formal analysis; Validation; Investigation; Methodology; Writing—original draft; Writing—review and editing. **Shaohe Wang**: Conceptualization; Formal analysis; Validation; Investigation; Methodology. **Yan Liu**: Investigation. **Karen F Oegema**: Conceptualization; Data curation; Formal analysis; Supervision; Funding acquisition; Validation; Project administration; Writing—review and editing. **Kevin F O'Connell**: Supervision; Funding acquisition; Validation; Writing—original draft; Project administration; Writing—review and editing.

Source data underlying figure panels in this paper may have individual authorship assigned. Where available, figure panel/source data authorship is listed in the following database record: biostudies:S-SCDT-10_1038-S44319-024-00157-y.

## Funding

## Disclosure and competing interests statement

The authors declare no competing interests.

# Expanded View Figures

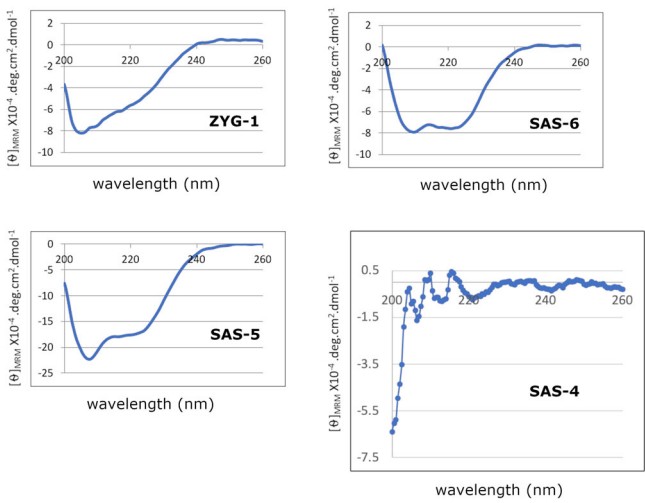

**Figure EV1.  Purification and characterization of recombinant core centriolar proteins.**

Circular dichroism spectra of the refolded proteins indicate secondary structural elements are present in the refolded proteins. The spectrum of ZYG-1 indicates that the protein consists of both alpha-helices and beta-sheets, while the spectra of SAS-5 and SAS-6 indicate that the proteins consist predominantly of alpha-helices (dual peak minima at wavelengths around 208 and 222 nm). SAS-4 appears to lack secondary structural elements.

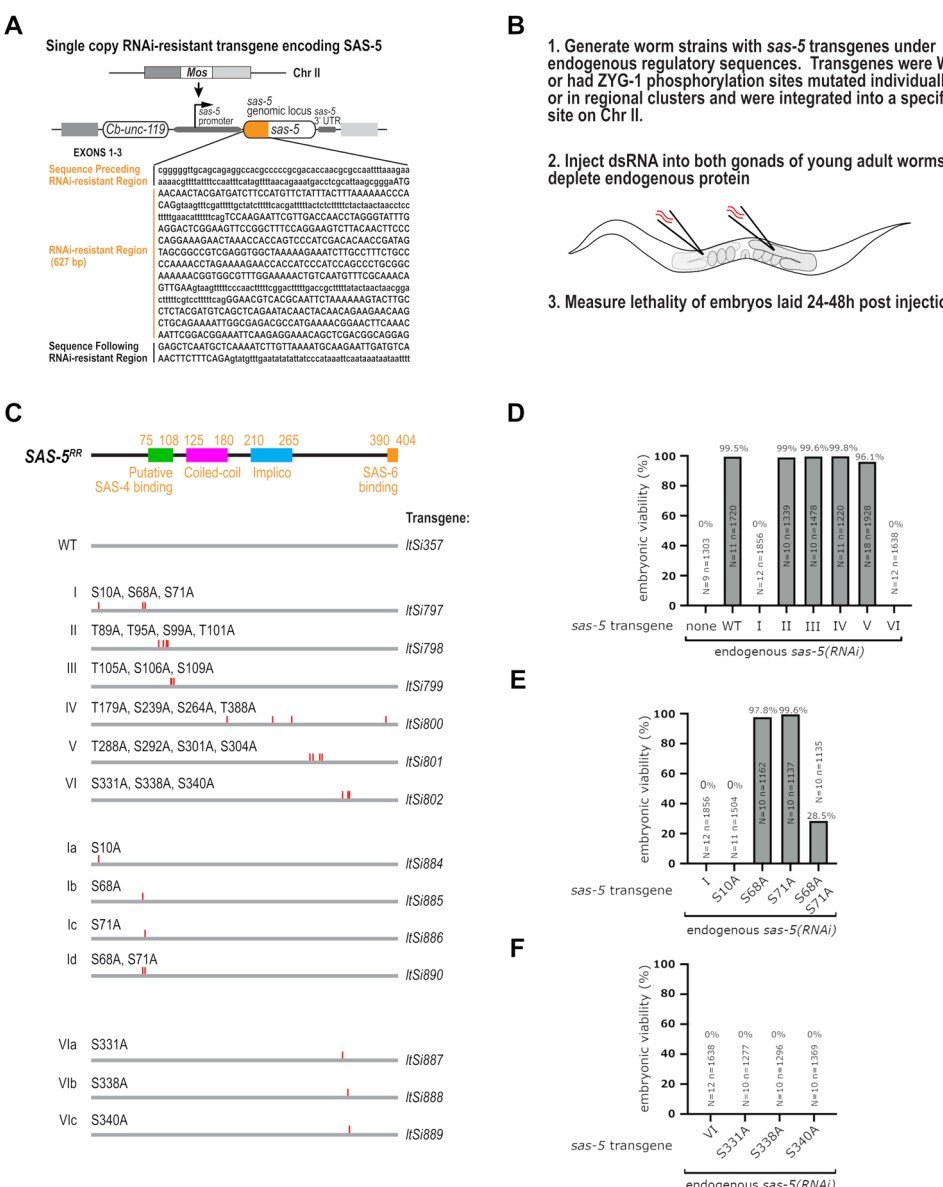

**Figure EV2. Screen of phospho-acceptor sites using mutant transgenes.**

(A) Schematic of the *sas-5* transgene integrated on chromosome II. The recoded portion of the transgene is indicated and allowed us to use RNAi to specifically target either the endogenous gene or the transgene itself. (B) Scheme used to assay embryonic lethality of control and mutant transgenes. (C) The initial screen utilized 6 mutant transgenes (I–VI), each comprising multiple serine-to-alanine mutations. The identity of the residues targeted in each construct is shown. (D–F) Embryonic viability of strains carrying the indicated versions of the *sas-5* transgene and subjected to *sas-5(RNAi)*. N=number of hermaphrodites and n = number of embryos counted. Note that the only single mutants that result in a complete loss of viability are S10A, S331A, S338A, and S340A.

**A**

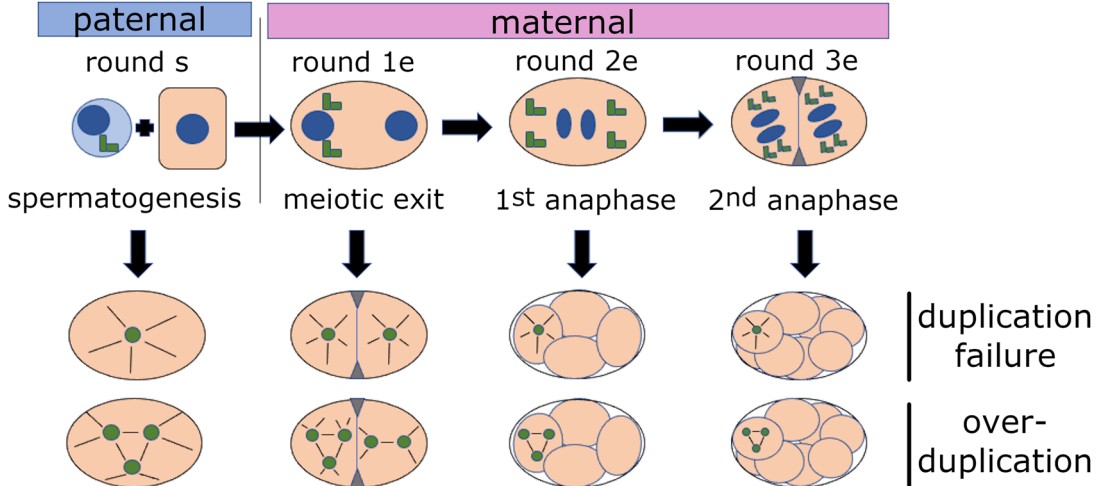

**B**

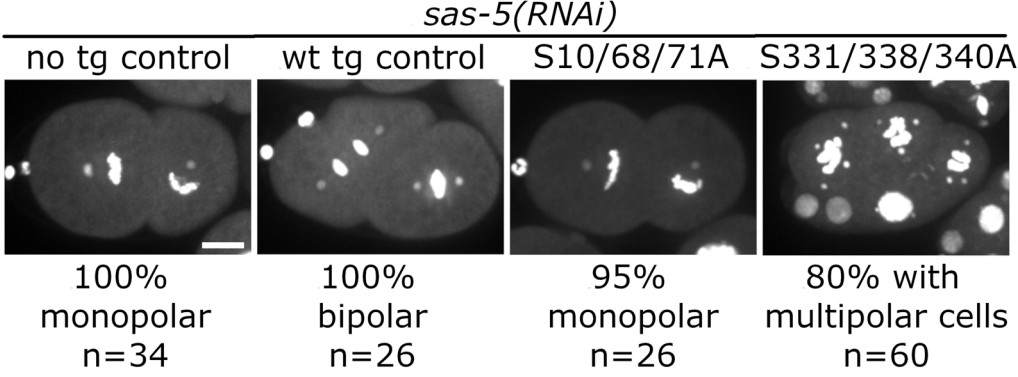

**C**

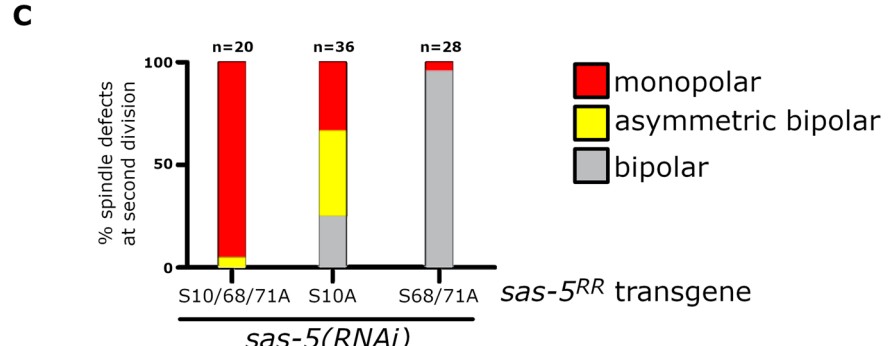

◀ **Figure EV3.  Centriole duplication errors in strains expressing mutant sas-5 transgenes.**

(A) An explanation of how errors in centriole duplication (duplication failure or overduplication) manifest as spindle assembly defects. The effect of these errors on spindle assembly are not immediately apparent as the mother and daughter centrioles remain in close association until the ensuing cell cycle when they separate in preparation for the next round of spindle assembly. This has special consequences for the early embryo where the first pair of centrioles are inherited exclusively from the sperm, while the components required for duplication are provided maternally. Hence paternal defects that occur during spermatogenesis (round s) lead too few or too many sperm centrioles and ultimately mono- and multipolar spindles in the one-cell embryo. In contrast maternal defects in centriole assembly which can occur during the first (e1), second (e2), or third round (e3 and so forth), result in spindle defects during the two-, four-, and eight-cell stages respectively. Depending on the severity of the defect, mono- or multipolar spindles might be observed in some or all of the cells of the embryo. For simplicity, spindle defects are only depicted in one of the cells of four- and eight-cell stage embryos. (B) Spindle assembly defects in strains expressing wild-type and mutant versions of the recoded *sas-5* transgene. All strains were subject to *sas-5(RNAi)* targeting the endogenous gene. The no transgene control possesses all monopolar spindles at the two-cell stage due a complete block in the first (1e) round of centriole assembly. In contrast the strain with a wild-type copy of the transgene only possesses bipolar spindles demonstrating the ability of the transgene to escape RNAi-based silencing. The strain expressing the S10/68/71A triple mutant exhibits a nearly complete block in centriole assembly leading to mostly monopolar spindles at the two-cell stage. The strain expressing the S331/338/340 A triple mutant exhibits a high frequency of multipolar spindles indicating overduplication occurred during the early embryonic divisions. The S331A, S338A and S340A single mutants all exhibited a low frequency of multipolar spindles. Bar, 10 μm. (C) Quantification of spindle defects in S10/68/71A, S10A, and S68/71A expressing strains. Asymmetric spindles contain one normal sized pole and a second much smaller pole. Such a defect has been attributed to a partial block in centriole assembly.

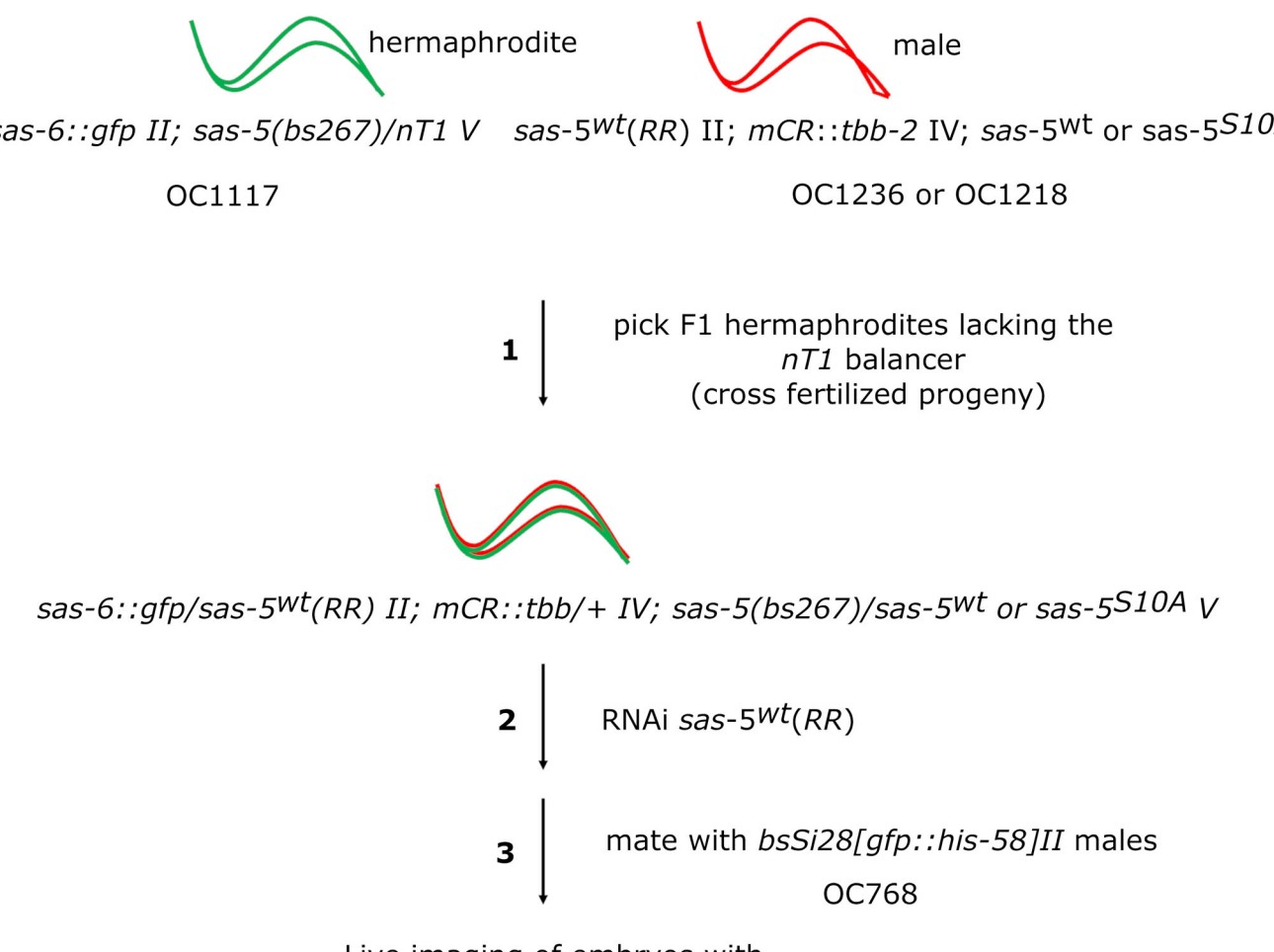

Figure EV4.  Genetic Scheme for SAS-6 recruitment assay.

The SAS-6 recruitment assay was performed with hermaphrodites carrying both the *sas-6::gfp* and rescuing recoded *sas-5^WT^(RR)* transgenes (red and green worm). Because these are both integrated at the same position on chromosome II we had to use following strategy to construct the desired hermaphrodites. Hermaphrodites of strain OC1117, carrying the *sas-6::gfp* transgene and heterozygous for the *sas-5* null allele *bs267* were crossed to males carrying the *sas-5^wt^(RR)* rescuing transgene and an mCherry (mCR) ::tubulin transgene (1). These males were either wild-type at the endogenous *sas-5* locus (strain OC1236) or carried the *sas-5^S10A^* mutation (strain OC1218). The resulting hermaphrodite cross progeny were treated with RNAi targeting the *sas-5^wt^(RR)* transgene (2) and mated to males of strain OC768 which expressed GFP::histone (3). Zygotes expressing GFP positive paternal chromatin were imaged.

