## [Peer Review File · EMBO Reports]

The Kinase ZYG-1 Phosphorylates the Cartwheel Protein SAS-5 to Drive Centriole Assembly in *C. elegans*

Kevin O'Connell, Prabhu Sankaralingam, Shaohe Wang, Yan Liu, and Karen Oegema

Corresponding author(s): Kevin O'Connell (kevino@intra.niddk.nih.gov), Prabhu Sankaralingam (prabhu.sankaralingam@nih.gov)

Review Timeline:

Submission Date:	11th Jan 24
Editorial Decision:	16th Feb 24
Revision Received:	29th Feb 24
Editorial Decision:	5th Apr 24
Revision Received:	5th Apr 24
Accepted:	22nd Apr 24

Editor: Deniz Senyilmaz Tiebe

Transaction Report:

Dear Dr. O'Connell,

Thank you for transferring your research manuscript to our journal, which was now seen by three referees, whose reports are copied below.

Referees express interest in the proposed role of ZYG-1 in regulation of centriole assembly. However, they also raise significant concerns that need to be addressed to consider publication here.

Given these positive recommendations, we would like to invite you to submit a revised manuscript. Please revise your manuscript with the understanding that the referee concerns (as in their reports) must be fully addressed and their suggestions taken on board. Please address all referee concerns in a complete point-by-point response. Acceptance of the manuscript will depend on a positive outcome of a second round of review. It is EMBO reports policy to allow a single round of major experimental revision only and acceptance or rejection of the manuscript will therefore depend on the completeness of your responses included in the next, final version of the manuscript.

We realize that it is difficult to revise to a specific deadline. In the interest of protecting the conceptual advance provided by the work, we recommend a revision within 3 months. Please discuss the revision progress ahead of this time with me if you require more time to complete the revisions, or if you have questions or comments regarding the revision (also by video chat).

1. A data availability section providing access to data deposited in public databases is missing (where applicable).
2. Your manuscript contains statistics and error bars based on $n=2$. Please use scatter plots in these cases.

You can submit the revision either as a Scientific Report or as a Research Article. For Scientific Reports, the revised manuscript can contain up to 5 main figures and 5 Expanded View figures, and it should not exceed 27000 characters. If the revision leads to a manuscript with more than 5 main figures it will be published as a Research Article. In this case the Results and Discussion section should be separate. If a Scientific Report is submitted, these sections have to be combined. This will help to shorten the manuscript text by eliminating some redundancy that is inevitable when discussing the same experiments twice. In either case, all materials and methods should be included in the main manuscript file.

4) a .docx formatted letter INCLUDING the reviewers' reports and your detailed point-by-point responses to their comments. As part of the EMBO publication's Transparent Editorial Process, EMBO reports publishes online a Review Process File (RPF) to accompany accepted manuscripts. This File will be published in conjunction with your paper and will include the referee reports, your point-by-point response and all pertinent correspondence relating to the manuscript.

<https://www.embopress.org/page/journal/14693178/authorguide#transparentprocess>

5) a complete author checklist, which you can download from our author guidelines

<https://www.embopress.org/page/journal/14693178/authorguide>. Please insert information in the checklist that is also reflected in the manuscript. The completed author checklist will also be part of the RPF.

6) Please note that all corresponding authors are required to supply an ORCID ID for their name upon submission of a revised manuscript (). Please find instructions on how to link your ORCID ID to your account in our manuscript tracking system in our Author guidelines

7) Before submitting your revision, primary datasets produced in this study need to be deposited in an appropriate public database (see <https://www.embopress.org/page/journal/14693178/authorguide#datadeposition>). Please remember to provide a reviewer password if the datasets are not yet public. The accession numbers and database should be listed in a formal "Data Availability" section placed after Materials & Method (see also

<https://www.embopress.org/page/journal/14693178/authorguide#datadeposition>). Please note that the Data Availability Section is restricted to new primary data that are part of this study. * Note - All links should resolve to a page where the data can be accessed. *

Additional information on source data and instruction on how to label the files are available:

<https://www.embopress.org/page/journal/14693178/authorguide#sourcedata>

9) Our journal encourages inclusion of *data citations in the reference list* to directly cite datasets that were re-used and obtained from public databases. Data citations in the article text are distinct from normal bibliographical citations and should directly link to the database records from which the data can be accessed. In the main text, data citations are formatted as follows: "Data ref: Smith et al, 2001" or "Data ref: NCBI Sequence Read Archive PRJNA342805, 2017". In the Reference list, data citations must be labeled with "[DATASET]". A data reference must provide the database name, accession number/identifiers and a resolvable link to the landing page from which the data can be accessed at the end of the reference. Further instructions are available at <http://www.embopress.org/page/journal/14693178/authorguide#referencesformat>

10) Regarding data quantification (see Figure Legends:

<https://www.embopress.org/page/journal/14693178/authorguide#figureformat>)

- the name of the statistical test used to generate error bars and P values,

- the number (n) of independent experiments (please specify technical or biological replicates) underlying each data point,

- the nature of the bars and error bars (s.d., s.e.m.),

- If the data are obtained from n Program fragment delivered error ``Can't locate object method "less" via package "than" (perhaps you forgot to load "than"? at //ejpvfs23/sites23b/embor_www/letters/embor_decision_revise_and_review.txt line 56.' 2, use scatter blots showing the individual data points.

12) Please also note our reference format:

I look forward to seeing a revised version of your manuscript when it is ready. Please let me know if you have questions or comments regarding the revision.

Kind regards,

Deniz Senyilmaz Tiebe

Deniz Senyilmaz Tiebe, PhD
Scientific Editor
EMBO Reports

Referee #1:

This is a neat and elegant study that provides new insight into the control of centriole duplication in *C. elegans*, and more specifically into molecular interactions of the conserved ZYG-1 kinase and the centriolar protein SAS-5/STIL. Centrosomal proteins are notoriously difficult to work with in vitro and as the authors correctly point out studies to date have mostly used protein fragments to identify protein-protein interaction domains and PLK4/ZYG-1 phosphorylation sites. To overcome this limitation, here the authors purified centriole duplication factors from bacterial inclusion bodies, refolded these and then confirmed native conformation/functionality of refolded proteins with circular dichroism assays. This methodology enabled successful expression of full-length SAS-4, SAS-5, SAS-6 and ZYG-1 proteins. Of these proteins, ZYG-1 phosphorylated SAS-5 the most, with over 30 sites revealed with phospho-mapping by mass-spec. The authors assayed the physiological relevance of several phosphosites in *C. elegans* development identifying the conserved residue Ser-10 in SAS-5 as the critical ZYG-1 target site. They found that mutating this residue prevented centriole assembly giving rise to monopolar spindles in two-cell embryos. Using live imaging the authors also demonstrated that Ser10 is required for stable incorporation of SAS-5-SAS-6 into procentrioles, and (directly or indirectly) also for subsequent incorporation of SAS-4.

The manuscript is interesting, clearly presented with high quality data, and as such it would appeal to the readership of EMBO Reports. I have listed a few points below the authors should address prior to publication.

1. The weakest aspect of the paper is the binding assays shown in Fig. 1. There appears to be considerable background binding to the GST tag, therefore the authors should mention this caveat when making comparisons between species in their discussion. While it is not critical for the conclusions of the paper, I still wonder why GST pulls down so much ZYG-1 in Fig 1E. I understand that binding in 1F is calculated relative to GST but it is nonetheless concerning that there is such a strong interaction with GST alone. Likewise, by eye, aside from 1-250KD there appears to be only a small difference between binding affinity of different GST-ZYG-1 fragments and GST alone for SAS-5.
2. If text is deemed too long, I would recommend shortening the Discussion because there is some overlap with the Results.
3. Please define what *ItSi357* transgene is in main text (currently it is defined only in the legend of Fig 2).
4. Main text states that "Surprisingly, we found that when subjected to either control or *sas-5RR* RNAi, the wild-type strain expressed similar levels of SAS-5 protein" Why is this surprising? This strain does not have a transgene, so *sas-5RR* has nothing to target. Or is this a wild-type strain that carries *ItSi357*? If so, this should be made clear.
5. For Fig 2A-G and Fig 5B and C, it would be helpful to show which labelled proteins are visualised (as done in Fig 4 panels).

Also, consider changing the colour of arrowhead depicting histones/DNA (currently red) as based on the fluorophores shown one would expect red to correspond to tubulin.

6. Fig 1D, check labelling as bottom two GST-SAS-5 constructs are both described as (180-404)

7. Fig 4A: typo, it should be 'marks' instead of 'markes'

8. In the merged document, EV3 appeared oversized and my pdf viewer could not visualise it.

Referee #2:

This manuscript by Sankaralingam and colleagues focuses on the molecular mechanism of centriole assembly in the *C. elegans* embryo. Despite numerous studies in the field, the target of ZYG-1 was not very clear. Here, the authors reveal that the phosphorylation of SAS-5 by ZYG-1 is crucial in the centriole assembly mechanism. The authors first demonstrate that SAS-5 is phosphorylated by ZYG-1 in vitro by successfully purifying soluble and functional proteins using a refolding approach. They identify several phosphorylated residues of SAS-5, particularly serine 10. Through in vitro and in vivo experiments, the authors show that the S10A mutation disrupts the stable incorporation of the SAS-5-SAS-6 complex, as well as the recruitment of SAS-4, suggesting that the phosphorylation of SAS-5 by ZYG-1 is critical for centriole assembly. The authors also identify three other phosphorylation sites on residues S331, S338, and S340 of SAS-5 by ZYG-1 that play a role in centriole amplification. The article is very clear, well-written, and the results are robust. It is an excellent work that distinctly identifies a specific SAS-5 residue whose phosphorylation by ZYG-1 is crucial for the assembly of the *C. elegans* centriole. This finding is significant for the field and enhances our understanding of the centriole duplication mechanism, shedding light on its evolution and divergence among different species (as discussed by the authors).

I therefore recommend this article for publication with just a few minor points:

- The authors assess the amount of protein in the different mutants by doing quantitative immunoblot. Does the total cell amount reflect the amount of protein at the centrosome? Would it be possible to assess the amount of protein localizing to the centrosome?

- in Figure 3G, the authors conclude that the centriole overduplicates. Would it be possible to do a centriole and PCM immunofluorescence labeling to verify that the centrosomes contain centrioles? It may be a fragmented PCM. It is possible that 3A is not responsible for overamplification but gives an unstable centriole which then fragments under the forces of the spindle and gives a fragmented PCM.

Referee #3:

Centriole biogenesis is an essential process and must be tightly regulated to avoid cell division errors and disease. While the core players have been identified, much remains to be understood about their unique interactions and enzymatic activity. In this manuscript, Sankaralingam et al use in vitro biochemistry and *C. elegans* as a model system to explore the kinase activity of ZYG-1 during centriole biogenesis. This work is an impressive biochemical undertaking to generate 4 notoriously difficult to purify centriolar proteins and validate proteins for proper folding and kinase activity. Based on phosphorylation sites determined from the in vitro assays, the authors use complementary RNAi and CRISPR approaches to determine the critical SAS5 phosphorylation sites. The combination of approaches was nice both for validation purposes and comparison of the two techniques. The authors find that serine 10 and serines 331/338/340 are required for SAS5 function in worms. Live imaging of worm development demonstrates that serine 10 functions in centriole assembly likely through stabilizing SAS6, while serines 331/338/340 function in bipolar spindle formation. Overall, this is a well-written manuscript with rigorous experiments, technical advancements, and well-founded conclusions. Minor comments follow.

Minor comments:

1. While the discussion touches on how ZYG-1 phosphorylation of SAS5 in worms compares to what is known for the homologs in human cells and *Drosophila*, it might be helpful to incorporate these similarities/differences into the model figure. This is especially relevant as we begin to understand nuances of centriole biogenesis that may be cell-type specific.

2. "Further as shown in Fig 1A, the in vitro refolded ZYG-1 exhibited enzymatic activity." I cannot find this data in figure 1A. Please clarify.

3. It would be helpful to point out α -helices and β -sheets in CD-spectrum in Fig EV1 or add more descriptive information to the figure legends.

February 29, 2024

Deniz Senyilmaz Tiebe, PhD
Scientific Editor
EMBO Reports

Dear Dr. Senyilmaz Tiebe,

We are pleased to submit our revised manuscript concerning the role of ZYG-1-mediated phosphorylation in centriole assembly. We thank the reviewers for their thoughtful comments and truly appreciate all three for taking time to carefully consider our work. Importantly, by addressing these concerns feel that we have substantially improved the manuscript. Below we address the reviewers' comments point by point and look forward to hearing from you.

Sincerely,

Kevin O'Connell
Senior Investigator
Laboratory of Biochemistry and Genetics,
NIDDK, National Institutes of Health, USA

Referee #1:

1. The weakest aspect of the paper is the binding assays shown in Fig. 1. There appears to be considerable background binding to the GST tag, therefore the authors should mention this caveat when making comparisons between species in their discussion. While it is not critical for the conclusions of the paper, I still wonder why GST pulls down so much ZYG-1 in Fig 1E. I understand that binding in 1F is calculated relative to GST but it is nonetheless concerning that there is such a strong interaction with GST alone. Likewise, by eye, aside from 1-250KD there appears to be only a small difference between binding affinity of different GST-ZYG-1 fragments and GST alone for SAS-5.

We agree with the reviewer. We ran our binding assay under many different conditions to try to eliminate background binding but were not able to do so. As requested, we have edited the text as shown below to alert the readers to this caveat.

Page 9

“We therefore performed *in vitro* pull-down experiments with GST-fusions of full-length and N-terminal truncations of SAS-5 as bait to capture ZYG-1 (prey) from solution. We found that full-length GST-SAS-5 could interact with ZYG-1 (Fig 1D-F). However, we failed to detect binding above background for all N-terminally truncated version of SAS-5, include one lacking just the first 107 amino acids. It is possible that these N-terminally truncated proteins bind to ZYG-1 with reduced efficiency, but that such weak binding is not detectable due to the relatively high background of our assay.”

Page 10

“In particular, the kinase domain (amino acids 1-250) showed the strongest binding. However, the L1 linker (250-350), the CPB (338-564), and PB3 (630-706) were also found to bind above background levels. While it remains to be seen if these later interactions contribute significantly to ZYG-1-SAS-5 binding, similar results have been obtained in vertebrates and flies (Cottee *et al.*, 2017; Kratz *et al.*, 2015; McLamarrah *et al.*, 2018; Ohta *et al.*, 2014)”

Page 26 (Discussion)

Here we show that ZYG-1 also physically associates with SAS-5 (Fig 1D-F). In light of this, it is interesting to note that ZYG-1 may have evolved a distinct mechanism for docking onto SAS-5. In flies and humans, Plk4 binds to the coiled-coil domain of STIL/Ana2 (Arquint *et al.*, 2015; Cottee *et al.*, 2017; McLamarrah *et al.*, 2018; Ohta *et al.*, 2014). While the limitations of our interaction assay leave open the possibility that ZYG-1 also binds the coiled-coil region of SAS-5, our results indicate that ZYG-1 makes important contacts outside of this region (Fig 1D-F).

2. If text is deemed too long, I would recommend shortening the Discussion because there is some overlap with the Results.

We appreciate this suggestion and if we need to shorten the manuscript prior to publication, we will work to shorten the discussion.

3. Please define what ItSi357 transgene is in main text (currently it is defined only in the legend of Fig 2).

We thank the reviewer for pointing out this oversight. We did describe the transgene on page 11 along with mutant versions but did not specifically point out that ItSi357 is the wild-type sas-5 transgene. We have corrected this error.

4. Main text states that "Surprisingly, we found that when subjected to either control or sas-5RR RNAi, the wild-type strain expressed similar levels of SAS-5 protein" Why is this surprising? This strain does not have a transgene, so sas-5RR has nothing to target. Or is this a wild-type strain that carries ItSi357? If so, this should be made clear.

We apologize for this error. This strain did in fact carry the ItSi357 transgene. We have edited the text to make this clear.

5. For Fig 2A-G and Fig 5B and C, it would be helpful to show which labelled proteins are visualised (as done in Fig 4 panels). Also, consider changing the colour of arrowhead depicting histones/DNA (currently red) as based on the fluorophores shown one would expect red to correspond to tubulin.

We believe the reviewer is referring to Figure 3A-G (and not 2A-G). We have added labels for the fluorescent proteins shown in Figures 3 and 5 as requested. We get the reviewers point about the arrowheads, but the legend does specify what each colored arrowhead is pointing to and to change the color of the arrowheads would require us to rebuild the entire figure, given the way the figure was constructed. We hope this is ok.

6. Fig 1D, check labelling as bottom two GST-SAS-5 constructs are both described as (180-404)

Thank you. Fixed.

7. Fig 4A: typo, it should be 'marks' instead of 'markes'

Fixed.

8. In the merged document, EV3 appeared oversized and my pdf viewer could not visualise it.

We are not sure why this occurred but will do our best to avoid this issue upon resubmission.

Referee #2:

- The authors assess the amount of protein in the different mutants by doing quantitative immunoblot. Does the total cell amount reflect the amount of protein at the centrosome? Would it be possible to assess the amount of protein localizing to the centrosome?

Unfortunately, our SAS-5 antibody does not work well in IF experiments and we did not want to epitope tag the mutant SAS-5 protein as we feared the tag might cause synthetic effects (that is having the S10A mutation and epitope tag on the same protein). We thought the better strategy would be to analyze SAS-6 since SAS-5 and SAS-6 form a complex and are mutually required for localization to centrioles. Thus SAS-6 can serve as a proxy for SAS-5. In doing this, we found for the S10A mutant, the initial levels of the SAS-5-SAS-6 complex as shown in Figure 4, are not different from those of the wild-type, but then decline due to a failure to stably incorporate the complex. Thus the S10A mutation does not affect the amount of the SAS-5-SAS-6 complex that initially localizes to the centrosome. For the 3A mutant, we measured the levels of SAS-6 at the centrosomes of early embryos but did not detect an increase relative to wild-type. This might be due to the rather modest increase in total levels (~2 fold increase) which would only be expected to result in a small (but evidently consequential) increase at centrioles that can't be reliably detected by quantitative IF. We have observed this before in certain mutants where SAS-6 is modestly overexpressed leading to extra centrioles but no apparent increase in centriole-associated SAS-6 levels (PMID: 35377871)

- in Figure 3G, the authors conclude that the centriole overduplicates. Would it be possible to do a centriole and PCM immunofluorescence labeling to verify that the centrosomes contain centrioles? It may be a fragmented PCM. It is possible that 3A is not responsible for overamplification but gives an unstable centriole which then fragments under the forces of the spindle and gives a fragmented PCM.

Thank you for this suggestion. We have stained 3A embryos for SAS-6 and tubulin and find that all spindle poles contain this centriole marker. The new data has been added to Fig 3. This indicates that all poles have a centriole and are not composed of an acentriolar fragment of PCM. While it is still possible that centrioles themselves are fragmenting in the 3A mutant, we think the most likely explanation for the extra spindle poles is that overexpression of SAS-5 in the 3A mutant is driving centriole overduplication. This seems very likely given that centrioles overduplicate when SAS-5/STIL is overexpressed in other systems (DOI: [10.1242/jcs.104109](https://doi.org/10.1242/jcs.104109) DOI: [10.1242/jcs.099887](https://doi.org/10.1242/jcs.099887) DOI: [10.1083/jcb.200910016](https://doi.org/10.1083/jcb.200910016))

Referee #3:

Minor comments:

1. While the discussion touches on how ZYG-1 phosphorylation of SAS5 in worms compares to what is known for the homologs in human cells and Drosophila, it might be helpful to incorporate these similarities/differences into the model figure. This is especially relevant as we begin to understand nuances of centriole biogenesis that may be cell-type specific.

We very much appreciate this suggestion and did contemplate adding this information. However our manuscript is presently on the long side and we are wary of incorporating any additional discussion that would further lengthen the paper. We hope the reviewer understands.

2. "Further as shown in Fig 1A, the in vitro refolded ZYG-1 exhibited enzymatic activity." I cannot find this data in figure 1A. Please clarify.

Sorry we referred to the wrong panel. Panel Fig. 1C. shows that ZYG-1 has enzymatic activity as it phosphorylates SAS-5. We have corrected this error.

3. It would be helpful to point out a-helices and b-sheets in CD-spectrum in Fig EV1 or add more descriptive information to the figure legends.

We thank the reviewer for pointing out the need for more information and as suggested we have added more descriptive information to the figure legend. We would like to emphasize however that CD spectra do not detect individual secondary structural elements such as alpha-helices and beta-sheets. Instead they detect the overall content of these secondary structural elements within a given protein. For instance, a protein composed entirely of alpha helix, exhibits dual peak minima at wavelengths close to 208 and 222 nm. A protein that is comprised fully of beta sheets, shows a single minimum centered around 215 nm. In reality, most proteins contain a mixture of the secondary structural elements. In such cases, the CD spectrum is an average of all of these elements.

The CD spectrum of our SAS-6 resembles a classical alpha helical protein and is consistent with the long coiled-coil helical tail which would be expected to dominate over the signal of beta sheets that are present in the head domain (based on the partial crystal structure of SAS-6). The spectrum of SAS-5 also shows heavy influence of alpha helices over other structural elements and is consistent with its known structure. In the case of ZYG-1, the peak minima are not clearly discernible, indicating a mixture of secondary structural elements in its backbone, which again is expected based on the known and predicted structural data. In summary, while the CD spectra can't reveal the exact conformation of our refolded proteins, they do show us that the alpha-helical and beta-sheet content of our proteins is consistent with their known or predicted structures, and importantly rule out the possibility that these proteins failed to refold.

Dear Kevin,

Thank you for submitting your revised manuscript. It has now been seen by two of the original referees.

As you can see, the referees find that the study is significantly improved during revision and recommend publication. However, I need you to address the points below before I can accept the manuscript.

- Data Availability section needs to be placed before Acknowledgements.
- Please remove the Author Contributions section from the manuscript.
- Please rename the Conflicts of Interests section as "Disclosure Statement and Competing Interests".
- We note that funding information is incomplete in the manuscript submission system - i.e. Intramural Research Program of the National Institutes of Health is missing.
- Please resubmit EV Tables as Dataset EV and rename them as Dataset EV1, 2 and 3 and update their in-text callouts accordingly. Please remove their legends from the manuscript text and include them in the file.
- The Data Availability section is reserved for the primary datasets generated in the study. If your study does not include datasets, please replace the current sentence with the following statement: This study includes no data deposited in external repositories.
- Our production/data editors have asked you to clarify several points in the figure legends:
 - o Please note that the legends for figures 3b-g is not provided in the sequential manner (legend for figure 3c, e, g is provided before legend of figure 3b, d, f). This needs to be rectified.
 - o Please note that information related to n is missing in the legend of figure 2e.
 - o Please note that the scale bar is missing for figures 4c; 5c.
 - o Please note that the scale bar needs to be defined for figure 3i.
 - o Please note that scale bar and its definition are missing for figure EV 3b.

Thank you again for giving us to consider your manuscript for EMBO Reports, I look forward to your minor revision.

Kind regards,

Deniz

--

Deniz Senyilmaz Tiebe, PhD
Scientific Editor
EMBO Reports

Referee #1:

The authors have satisfactorily address my comments, so I am pleased to recommend this article for publication.

Referee #2:

I would like to thank the authors for their answers and their work. In view of the results and the quality of the work, I recommend this article for publication.

Referee #3:

The authors have adequately addressed my comments. This is a nice manuscript that will advance our understanding of centriole biogenesis.

The authors have addressed all minor editorial requests.

Dear Kevin,

Thank you for submitting your revised manuscript. I have now looked at everything and all is fine. Therefore, I am very pleased to accept your manuscript for publication in EMBO Reports.

Congratulations on a nice work!

Kind regards,

Deniz

--

Deniz Senyilmaz Tiebe, PhD
Editor
EMBO Reports

--
